# Shape-memory polyurethanes for polar wearables with ultrasensitive multi-monitoring

Tianze Chen[1,2], Jing Xu [1] ✉, Chongyang Wang[1,2], Xinrui Zhang [1] ✉, Xianqiang Pei[1], Tingmei Wang[1,2] & Qihua Wang[1,2]

Integrating environmental stability and multi-monitoring modules into flexible sensor remains a pivotal scientific challenge. This study presents a supramolecular polyurethane (PU) engineered with fluorine-rich segments that form electrostatic crosslinks with positively charged ionic groups at polymer chain terminals and establish fluorine-dipole interactions with blended ionic liquid (IL) to stabilize ion transport pathways. The resulting ionically conductive elastomer combines shape memory capacity, self-healing property, and cryogenic tolerance, retaining robust mechanical strength (~32.31 MPa), toughness (~107.05 MJ m$^{-3}$) and substantial ionic conductivity even at −40 °C. Notably, it exhibits a high temperature coefficient of resistance (TCR = 8.05% °C$^{-1}$) at cryogenic temperatures (−40 °C to −30 °C), making it attractive for the development of cryogenic sensing materials. Additionally, the material exhibits high-sensitivity physiological monitoring capabilities with signal fidelity, serving as ionic skins for accurate physiological signal acquisition. Such multifunctional adaptability positions it as an ideal candidate for next-generation flexible electronics requiring reliable performance in extreme environments.

The rapid advancement of flexible electronics technology is driving the innovation of sensing systems toward stretchable, adaptive, and self-healing directions. Compared to traditional rigid sensors, flexible sensors demonstrate disruptive application potential in health monitoring, human–machine interaction, and soft robotics due to their conformal adhesion, mechanical durability, and biocompatibility[1–5]. The core development of flexible sensors lies in creating materials that exhibit environmental stability, self-healing capability, and high signal fidelity. Based on their transmission mechanisms, flexible sensors are categorized into two types, electronic sensors relying on electrical conductors and ionic sensors based on ion-conductive systems[6]. Electronic sensors face limitations, such as opacity, excessive substrate dependency, and high rigidity with low stretchability, hindering their direct application[7–13]. In contrast, ionic sensors inherently possess homogeneous conductive epidermal characteristics. Their emergence

has advanced wearable medical technologies and soft robotic sensing fields due to significant conductivity changes under strain or pressure stimuli[6,14–18]. Despite progress, conventional flexible sensors primarily depend on hydrogel or elastomer matrices infused with ionic liquids (ILs) or conductive fillers, struggling to simultaneously achieve high ionic conductivity, environmental stability, and shape memory performance critical for epidermal sensor applications[19,20]. For instance, hydrogel-based ionic conductors suffer from poor stability, rapidly swelling in humid environments or dehydrating in arid conditions[21–23]. Recent efforts to address these limitations focus on dynamic polymer networks.

Among diverse materials, polyurethanes (PUs) have been widely utilized due to its chemical stability and remarkable molecular designability[24–27]. The development of high-performance PUs with functional applications through rational molecular design has

[1]State Key Laboratory of Solid Lubrication, Lanzhou Institute of Chemical Physics, Chinese Academy of Sciences, Lanzhou, China. [2]Center of Materials Science and Optoelectronics Engineering, University of Chinese Academy of Sciences, Beijing, China. ✉e-mail: jingxu@licp.cas.cn; xruiz@licp.cas.cn

increasingly become a research focus[28–35]. Benefiting from the rapid progress in supramolecular polymers, the incorporation of dynamic interactions, such as hydrogen bonds, metal coordination bonds, and ionic bonds, can enable energy dissipation through bond rupture and reconfiguration while enhancing crosslinking density, thereby improving tensile strength and fracture elongation[36–41]. However, integrating self-healing and shape memory capabilities with environmentally stable ionic conduction in PUs remains a significant challenge, rooted in the inherent conflict between energy-dissipative network reconfiguration and continuous charge transport pathways. A promising strategy involves leveraging fluorine-mediated interactions to enhance both ionic transport and stretchability in PU systems. Concurrently, the introduction of ILs, particularly 1-ethyl-3-methylimidazolium bis(trifluoromethylsulfonyl) imide ([EMIM]$^+$[TFSI]$^-$), known for its low viscosity and broad electrochemical window—exploits fluorine's strong dipole–dipole interactions and chemical inertness, endowing PU/IL hybrid systems with high stability[42–45]. For instance, Chen et al. reported a fluorinated PU-based material blended with ILs, achieving an ionotronic skin with high ionic conductivity ($1.04 \times 10^{-3}$ S cm$^{-1}$) and exceptional underwater stability[46]. Nevertheless, existing studies predominantly focus on fluorine's dielectric or hydrophobic properties, overlooking its potential to synchronously optimize environmental stability, self-healing kinetics and shape memory abilities.

This study proposed a molecular engineering paradigm that leverages the dual functionality of fluorine to construct spatially correlated yet functionally independent dynamic networks within a PU matrix. By introducing 2,2'-bis(trifluoromethyl) benzidine (TFMB) as a fluorinated chain extender and malonic acid dihydrazide (MDH) as a multi-H-bonding unit, we established a hierarchical architecture that decouples energy-dissipative sacrificial bonds from anion-conductive pathways. The trifluoromethyl groups in TFMB exhibited dual roles, generating synergistic effects in the PU matrix, forming ionic bonds with the end-capping agent trimethylammonium bromide (TLB) via their high electronegativity while reinforcing the molecular network alongside hierarchical H-bonds to preserve topological integrity and stability. Concurrently, polarized fluorine centers created localized electrostatic traps that immobilize cations through directional dipole interactions. And the fluorine-mediated cation confinement constructed "anion highways" for rapid anion migration and conductive pathway establishment. Remarkably, even when these highways were narrowed under harsh conditions, including low temperatures or mechanical stretching, the fluorine-stabilized framework preserved continuous ion-conducting channels through persistent dipole effects, maintaining stable ion transport efficiency.

Experimental results demonstrated breakthrough synergistic performance. The ionically conductive elastomer combines high stretchability (>900%) and ionic conductivity ($1.34 \times 10^{-4}$ S cm$^{-1}$) with self-healing behavior (85.4% for 1 h at room temperature). Additionally, the material exhibits shape-memory performance (shape recovery rate up to 94.8%) and damping capacity. The developed flexible sensor demonstrated cryogenic tolerance, effectively addressing the critical challenge of performance degradation under extreme thermal conditions. Notably, even at −40 °C that most polymeric sensors undergo catastrophic embrittlement, it retained mechanical resilience, achieving a tensile strength of 32.31 MPa and toughness of 107.05 MJ m$^{-3}$. Furthermore, the fabricated sensor not only enabled precise temperature sensing with a high temperature coefficient of resistance (TCR) of 8.05%°C$^{-1}$, but also served as a strain sensor for comprehensive health monitoring applications. Most importantly, it achieved clinically relevant sensitivity in electrocardiogram (ECG) monitoring, with T-wave to R-wave amplitude ratios within the critical diagnostic threshold range of 0.2–0.4, meeting medical-grade cardiovascular assessment standards. As illustrated in Fig. 1a, this combination of properties enables applications in polar-environment sensing and cryotherapy. Our work provides insights into the design of durable ionic conductors and offers a polymer engineering strategy that mitigates conventional performance trade-offs.

## Results

### Polymer design, preparation, and characterization

The high electronegativity of fluorine enables effective immobilization of cationic moieties within the material matrix, facilitating the multifunctional transformation of conventional PUs through fluorine engineering. Our design strategy focuses on constructing a synergistic dynamic network via the integration of fluorine–cation interactions (ionic bonds) and high-density hydrogen bonding units in PU architecture. This approach simultaneously enhances mechanical strength while leveraging the strong electrostatic nature of C–F bonds to amplify electron polarization effects. The resultant electronic polarization promotes cation immobilization while creating additional free volume for mobile ions, thereby improving ionic conductivity. As illustrated in Supplementary Figs. 1 and 2, a series of poly(urea-urethane) (PUU) and PUFT$_x$ were synthesized via a two- or four-step process. PUU referred to the control sample without both TFMB and TLB, whereas fluorinated counterparts were designated as PUFT$_x$, where $x$ denotes the millimolar ratio of TLB incorporation.

Within the PU network, MDH reacted with isocyanate groups to form acyl semi carbazide units rich in H-bond donors and acceptors. Concurrently, ionic interactions between TFMB and TLB established a collaborative dynamic crosslinking network. This dual dynamic network architecture enabled the optimized PUFT$_2$ sample to achieve exceptional mechanical properties with a tensile strength of 86.3 MPa and toughness of 311.5 MJ m$^{-3}$ (Supplementary Fig. 3). To develop multifunctional ionic elastomers, [EMIM]$^+$ [TFSI]$^-$ was incorporated into PUU and PUFT matrices, designated as PUFT$_x$-$y$IL (PUU-$y$IL when $x = 0$), where $y$ represents the mass fraction of [EMIM]$^+$ [TFSI]$^-$. The [EMIM]$^+$ cations preferentially interacted with fluorinated TFMB segments through fluorine–cation interactions, while [TFSI]$^-$ anions remained mobile to establish continuous conductive pathways (Fig. 1b). Systematic modulation of [EMIM]$^+$ [TFSI]$^-$ content ($y$) allowed precise control over free ion concentration and ionic conductivity. Consequently, these fluorine-engineered PU ionic elastomers with synergistic dynamic networks enabled a balanced integration of mechanical robustness and electrical functionality.

Fourier-transform infrared (FT-IR) spectra of all PUFT$_x$ samples (Supplementary Fig. 4) confirmed successful PU synthesis through the absence of characteristic isocyanate peaks between 2260 and 2280 cm$^{-1}$[47]. Simultaneously, the $^1$H NMR results (Supplementary Figs. 5 and 6) also confirmed the successful synthesis of PU and the effective incorporation of [EMIM]$^+$ [TFSI]$^-$. Given the structural similarity across PUFT$_x$ variants and the optimal mechanical performance of PUFT$_2$, we selected PUFT$_2$-$y$IL as the primary model system for generalized conclusions. Thermal gravimetric analysis (TGA) curves in Supplementary Fig. 7 demonstrated thermal stability for all PUFT$_2$-$y$IL elastomers, with the thermal decomposition temperatures ($T_d$, temperature of 5% weight loss) exceeding 255 °C, attributable to the inherent thermal resistance of [EMIM]$^+$ [TFSI]$^-$. Differential scanning calorimetry (DSC) revealed depressed glass transition temperatures ($T_g$) with increasing IL content (Fig. 2a), arising from IL-induced effects that reduced intermolecular interactions and increased free volume. Notably, the lowered $T_g$ facilitated rapid chain mobility at ambient or low temperatures, enabling dynamic noncovalent bond reconfiguration and conferring room-temperature self-healing capabilities with retained low-temperature tolerance. The absence of crystallization-related endothermic and exothermic peaks in DSC profiles, coupled with the results of X-ray diffraction (XRD) analysis (Fig. 2b), confirmed the noncrystalline nature of these ionic elastomers.

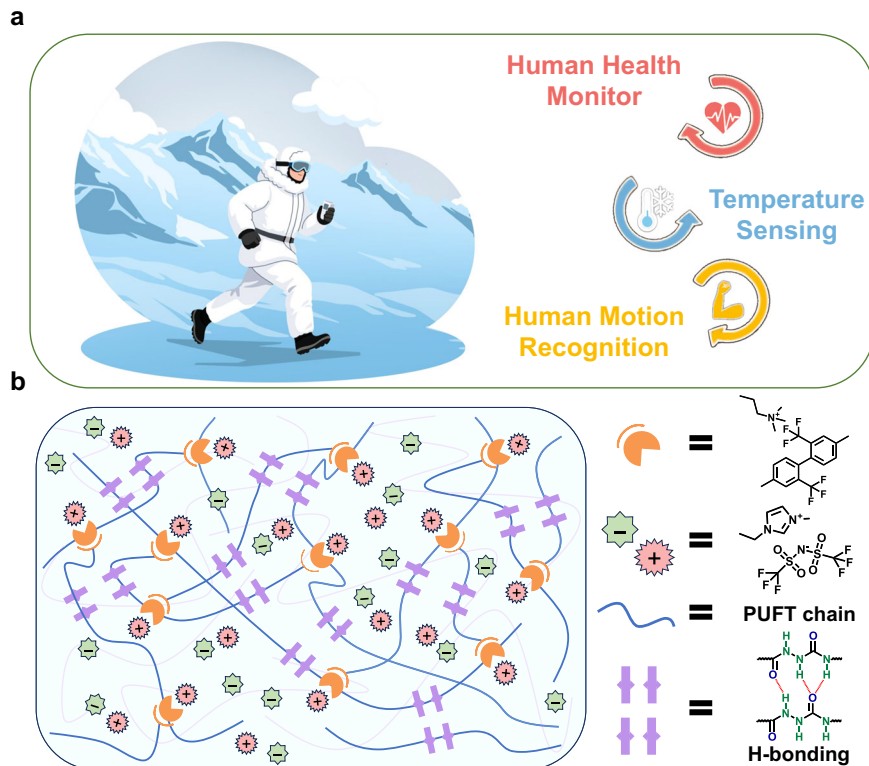

**Fig. 1 | Material design and functional implementation. a** Schematic for applications in polar environments. **b** Schematics and molecular formula of PUFT$_x$-$y$IL showing electric interactions, fluorine–cation interactions and H-bonds between each component.

## Validation of fluorine–cation interaction

The fluorine–cation dipole interaction constitutes the core of this molecular design strategy, playing dual critical roles in stabilizing the dynamic network and enabling anion-selective conduction. However, the existence patterns and interaction intensities of such interactions within multicomponent PU systems remain debated. To clarify the fluorine-mediated cation immobilization mechanism, we employed a multimodal characterization approach integrating quantum chemical calculations, spectroscopic and energy spectrum analyses, and surface ion distribution imaging. This systematic investigation not only decouples the contributions of fluorine's electronegativity and steric effects but also establishes quantitative correlations between interaction energy and macroscopic performance.

Quantum chemical calculations were first conducted to evaluate the binding energy of [EMIM]$^+$ [TFSI]$^-$ in the presence and absence of polymer chains with the computational details provided in Supplementary Information. As shown in Fig. 2c, the binding energy of [EMIM]$^+$ [TFSI]$^-$ under polymer chain influence was significantly more negative compared to its isolated state. Given that binding energy values were typically negative with more negative values indicating stronger binding and structural stability, this result suggested that the cations and anions of the IL readily dissociate via fluorine–cation interactions without disrupting the ionic bonding between molecular chains. This separation mechanism ensured the integrity of the dynamic network while facilitating ion mobility.

Structural validation of fluorine–cation interactions was further pursued through spectroscopic techniques. Supplementary Fig. 8 and Fig. 2d presented the FT-IR spectra of PUFT$_2$-$y$IL in the 600–3600 and 3050–3200 cm$^{-1}$ regions. Within the 3050–3200 cm$^{-1}$ range, two characteristic C–H stretching vibration peaks were observed, corresponding to the imidazolium ring of [EMIM]$^+$ cations. Four distinct vibrational modes were resolved through spectral deconvolution, consisting of asymmetric C (4)-H and C (5)-H (double-bonded carbon

atoms in imidazole ring) stretching (fit peek 1) as well as symmetric C–H stretching (fit peek 2), asymmetric C (2)-H stretching between nitrogen atoms (fit peek 3), and CH$_3$(N)HCH sidechain stretching (fit peek 4)[46,48]. Notably, fit peaks 3 and 4 in PUFT$_2$-50% IL exhibited blue shifts compared to PUU-50%IL, attributable to enhanced electron-withdrawing effects at C (2)-H sites through fluorine's induction effect. Conversely, fit peaks 1 and 2 displayed red shifts, indicating reduced constraints on C (4) and C (5) positions distal to polymer chains. Additionally, the C–F stretching vibration peaks in the 1180–1200 cm$^{-1}$ range exhibited pronounced intensity enhancement and broadening upon IL incorporation, confirming the dominance of [TFSI]$^-$-derived C–F vibrations and verifying successful IL integration. Complementing FT-IR analysis, Raman spectroscopy, with its sensitivity to symmetric vibrational modes and polarizability changes, provides further insights into fluorine-mediated structural reorganization. Supplementary Fig. 9 illustrated the characteristic [EMIM]$^+$ bands in the 1300–1600 cm$^{-1}$ region. For PUFT$_2$-50% IL, the H–C–H rock vibration at 1458 cm$^{-1}$ and scissor vibration at 1482 cm$^{-1}$ shifted to 1473 and 1495 cm$^{-1}$, respectively. These blue shifts reflected restricted bond vibrations of [EMIM]$^+$ cations, directly evidencing their attraction to the fluorine-rich PUFT matrix[46,48].

X-ray photoelectron spectroscopy (XPS) was employed to resolve fluorine's chemical states and quantify its binding with cationic species. The F 1$s$ spectra of PUFT$_2$, PUU-50%IL, and PUFT$_2$-50% IL displayed a single peak corresponding to fluorine in either polymer backbones or [TFSI]$^-$ anions (Supplementary Fig. 10). Notably, the C (–CF3) binding energy in PUFT$_2$ increased by 0.3 eV upon IL incorporation (Fig. 2e), indicating reduced electron cloud density on carbon atoms within trifluoromethyl groups. This reduction stemmed from fluorine's strong electronegativity and its interactions with [EMIM]$^+$ cations. Concurrently, F 1$s$ peak's binding energy shifted from 687.3 to 688.3 eV showing a 1.0 eV energy gap (Fig. 2f), further corroborating interactions between polymer chains and [EMIM]$^+$. To spatially resolve

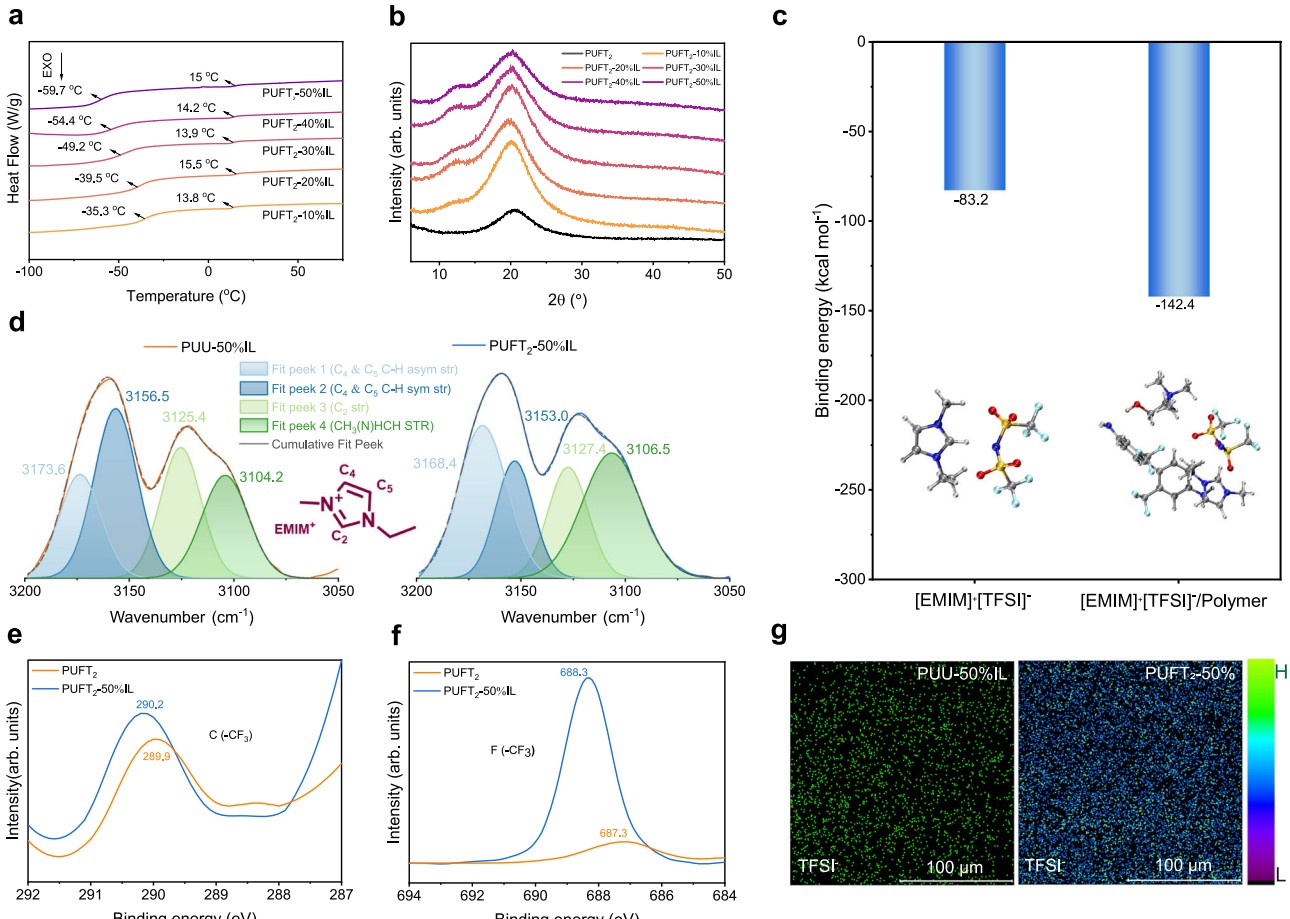

**Fig. 2 | Polymer characterizations and fluorine–cation interaction verification.** **a** DSC thermos-grams of PUFT$_2$-$y$IL. **b** XRD patterns of PUFT$_2$-$y$IL. **c** Calculated binding energies for two systems. The inset schematic shows the molecular models of the [EMIM]$^+$ [TFSI]$^-$ ion pairs and the [EMIM]$^+$ [TFSI]$^-$/polymer chain segment complexes. **d** FT-IR spectra of [EMIM]$^+$ cations in PUU-50%IL and PUFT$_2$-50% IL. The origin peak is divided into four fit peaks. **e**, **f** XPS spectra of carbon atoms and fluorine atoms in PUFT$_2$ and PUFT$_2$-50% IL. **g** ToF-SIMS pictures of PUU-50%IL and PUFT$_2$-50% IL.

fluorine distribution and track cation-anion coordination at the molecular scale, time-of-flight secondary ion mass spectrometry (ToF-SIMS) mapping was performed. Line-scan profiles of [TFSI]$^-$-related fragments (selected ion: CNO$_4$F$_3$S$_2^-$) revealed stark contrasts between matrices (Fig. 2g). PUFT$_2$-50% IL exhibited uniform brightness with minimal dark regions, whereas PUU-50%IL displayed pronounced brightness variations and extensive dark areas. This disparity confirmed that fluorine-mediated interactions promote homogeneous IL distribution within the PUFT$_2$ matrix, directly validating the spatial uniformity of fluorine–cation interactions.

Collectively, this multimodal characterization, including spanning quantum calculations, vibrational spectroscopy, surface chemistry analysis, and ion mapping unambiguously confirms the existence, intensity, and functional dominance of fluorine–cation interactions. These interactions not only stabilize the dynamic network but also govern anion-selective ion transport by spatially confining [EMIM]$^+$ cations while liberating [TFSI]$^-$ anions for conductive pathways. The decoupling of fluorine's electronegativity and steric effects provides a foundational framework for rational design of fluorine-engineered PUs with tunable mechanical and electrical properties.

### Mechanical and electrical properties

The mechanical properties of PUFT$_2$-$y$IL, including tensile strength, elongation at break, and toughness, were summarized in Supplementary Table 1, with corresponding stress–strain curves illustrated in

Fig. 3a. The introduction of IL immediately reduced material strength and toughness, likely due to IL–fluorine interactions disrupting original ionic bonds and weakening interchain cohesion. As IL content increased, mechanical strength and stretchability further decreased (Fig. 3b). Although PUFT$_2$ exhibited 86.3 MPa strength, the incorporation of 50% IL altered its strength and elongation to 1.32 MPa and 948%, respectively, achieving flexibility and mechanical compliance suitable for epidermal electronics. Small-angle X-ray scattering (SAXS) analysis revealed distinct scattering peaks for PUU (microphase separation period $d$ = 7.46 nm) and PUFT$_2$ ($d$ = 8.62 nm), confirming enhanced microphase separation due to ionic bonding between TFMB and TLB (Fig. 3c). The increased d-spacing in PUFT$_2$ indicated that ionic bonds amplify microphase-separated domains, positively influencing mechanical performance, as further corroborated by stronger halos in 2D-SAXS patterns compared to PUU (Supplementary Fig. 11). However, no scattering peaks were observed for PUU-50% IL or PUFT$_2$-50% IL, suggesting IL-induced disruption of microphase separation. Atomic force microscopy (AFM) images revealed progressively blurred boundaries between hard segments (bright regions) and soft segments (dark regions) in PUFT$_2$-$y$IL with increasing IL content (Supplementary Fig. 12), further evidencing weakened microphase separation.

Building on the mechanical adaptability of the fluorine-engineered dynamic network, we further investigated its electrical properties to elucidate how fluorine–cation interactions synergistically regulate ion transport mechanisms. Electrochemical impedance

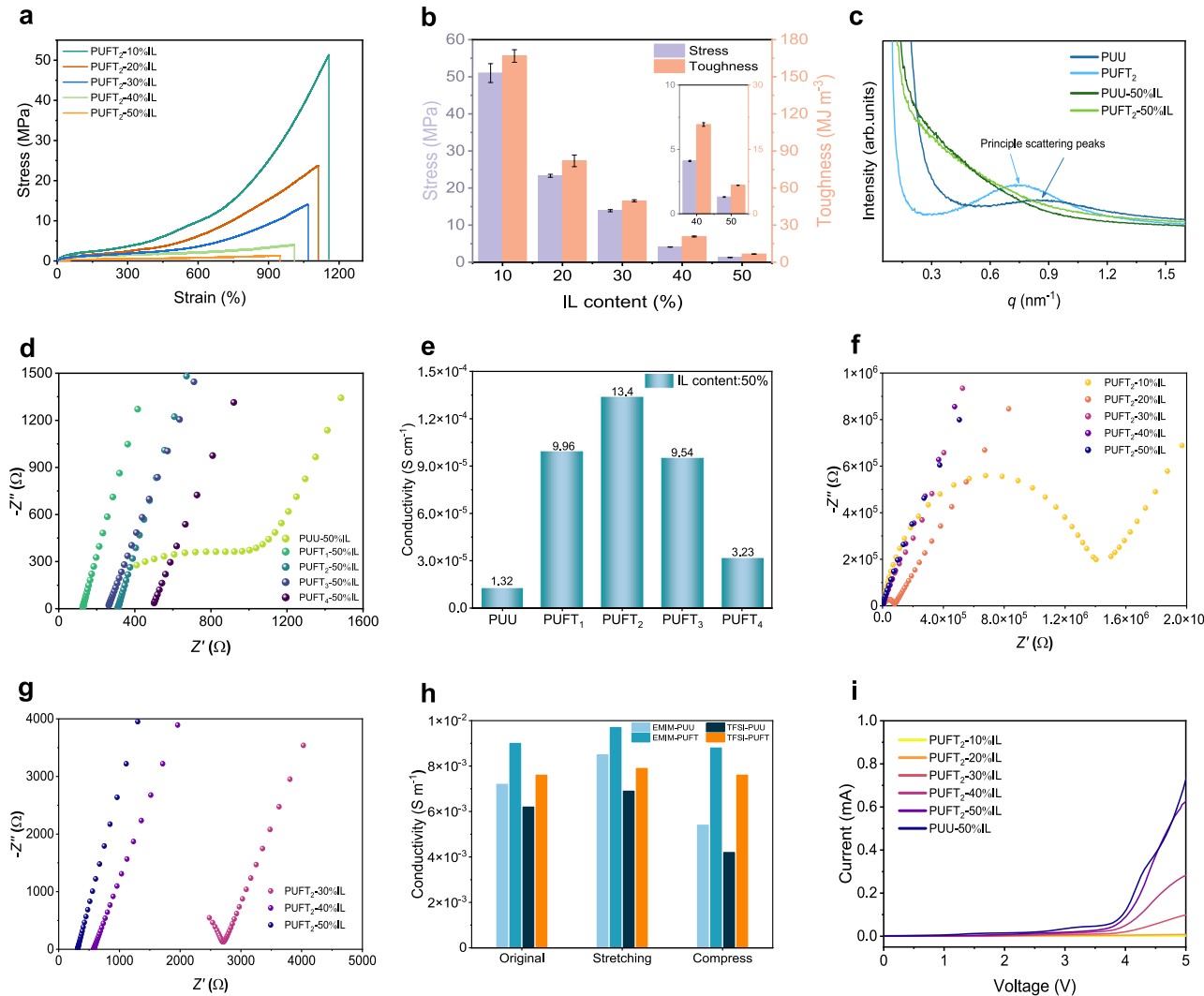

**Fig. 3 | Mechanical and electrical properties. a** Stress–strain curves of PUFT$_2$-$y$IL. **b** Tensile strengths and toughnesses of PUFT$_2$-$y$IL. Data are presented as mean values ± SD ($n$ = 3). **c** 1D-SAXS profiles of PUU, PUFT$_2$, PUU-50% IL and PUFT$_2$-50% IL. **d** Nyquist plot of PUU-50%IL and PUFT$_x$-50% IL. **e** Ionic conductivity of PUU-50%IL and PUFT$_x$-50% IL. **f, g** Nyquist plot and details of high-frequency area of PUU-50%IL and PUFT$_x$-50%IL. **h** Simulated ionic conductivity of PUU-IL and PUFT-IL systems. **i** LSV curves of PUU-50%IL and PUFT$_x$-50%IL.

spectroscopy (EIS) of PUU-50% IL and PUFT$_x$-50% IL (Fig. 3d) showed that bulk resistance initially increased and then decreased with the $x$ value changed. As shown in Fig. 3e, PUFT$_2$-50% IL achieved the highest ionic conductivity ($1.34 \times 10^{-4}$ S cm$^{-1}$), which was attributed to the fluorine–cation interactions that preferentially immobilized the [EMIM]$^+$ cations while weakening the anion–cation interaction, thereby promoting free ion release. Increasing IL concentration enhanced ionic conductivity (Fig. 3f, g and Supplementary Table 2), though concentrations exceeding 50% compromised mechanical stability and self-supporting capability, rendering them unsuitable for practical polymer electrolyte applications. Combined with mechanical data, this suggested that IL-induced destruction of polymer crystalline structures enhances amorphous domains, leading to mechanical degradation. The enhanced effect of fluorine engineering on ionic conductivity was further investigated by all-atom molecular dynamics simulations. Supplementary Figs. 13 and 14 illustrated the modeling of the polymer chain in the PUU-50%IL and PUFT-50%IL systems, with modeling and simulation details provided in the Supplementary Information. The results revealed superior conductivity in the fluorine-engineered PUFT-IL system compared to PUU during tensile-compression cycles (Fig. 3h). Notably, the PUFT-IL system

maintained stable conductivity during simulated epidermal sensor stretching-recovery processes, whereas PUU exhibited significant conductivity fluctuations, confirming fluorine engineering's critical role in ensuring sensing stability. Linear sweep voltammetry demonstrated electrochemical stability windows of 3.8–3.6 V for PUFT$_2$-$y$IL with 10–50% IL content, consistently exceeding the 2.7 V window of PUU-50%IL (Fig. 3i). The preserved high electrochemical stability, coupled with tunable ionic conductivity and mechanical compliance, establishes PUFT$_2$-$y$IL as a viable candidate for epidermal electronic devices.

## Self-healing, damping and shape memory properties

Driven by reversible intermolecular multi-hydrogen bonding and ionic interactions, the PUFT2-50% IL ionic elastomer exhibits self-healing behavior. Real-time resistance measurements during multiple surgical blade cuts showed that the material restored its original resistance within 0.5 s, confirming rapid electrical recovery (Fig. 4a). To quantify self-healing efficiency, PUFT$_2$-50% IL samples were cut into two pieces, then the two completely separated ruptured surfaces were brought into full contact at room temperature. As shown in Fig. 4b, tensile strength and strain gradually increased with healing time. After 1 h, the

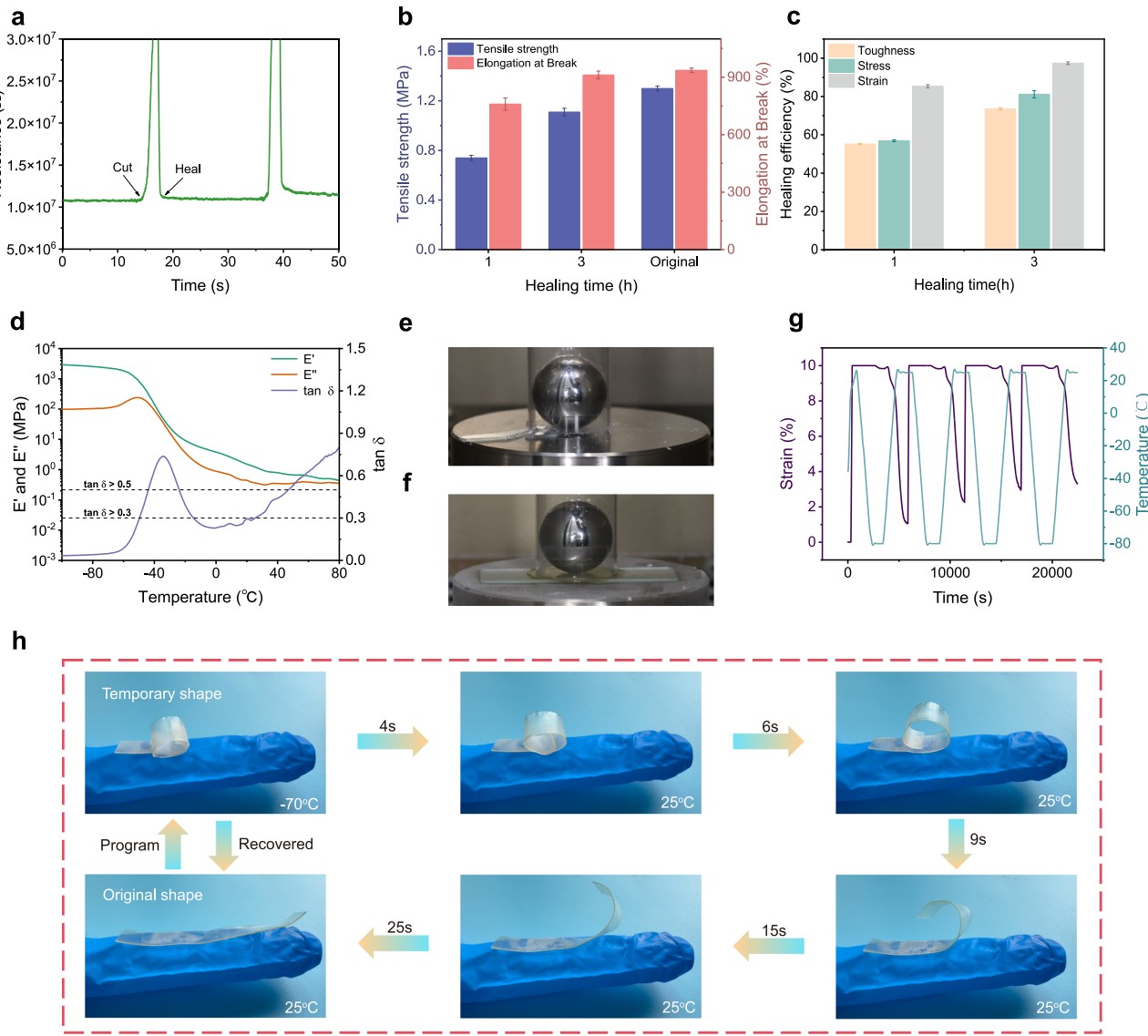

**Fig. 4 | Self-healing, damping and shape memory properties. a** Real-time resistance changes of PUFT$_2$-50% IL during cut-offs and healing. **b** Tensile strength and elongation at break of PUFT$_2$-50% IL at different healing times. Data are presented as mean values +/− SD ($n = 3$). **c** Healing efficiency of PUFT$_2$-50% IL at different healing times. Data are presented as mean values ± SD ($n = 3$).

**d** Temperature-dependent DMA results of PUFT$_2$-50% IL. **e** Fracture of unprotected glass from a 100 g ball falling from 50 cm height. **f** Glass protected by PUFT$_2$-50% IL from a 100 g ball falling from 50 cm height. **g** Shape memory curves of PUFT$_2$-50% IL. **h** Illustration of the temperature-responsive shape memory effect of PUFT$_2$-50% IL.

tensile strength and strain recovered to 0.74 MPa and 760%, respectively. After 3 h, they reached 1.11 MPa and 912%, similar to the original sample. For strain sensors, stretchability is a critical parameter. Based on strain recovery, the healing efficiency reached 85.4% after 1 h and 97.4% after 3 h (Fig. 4c), demonstrating its efficient self-healing capability.

For sensor materials, exceptional damping performance is essential. Tan $\delta$ is a key factor for evaluating damping properties. A temperature range with tan $\delta$ greater than 0.3 is considered an effective damping temperature range, while tan $\delta$ above 0.5 indicates a broad damping temperature range[49–52]. The temperature-dependent damping properties of PUFT$_2$-50% IL were measured via dynamic mechanical analysis (DMA) at 1 Hz (Fig. 4a). The results showed that tan $\delta$ remained above 0.3 from −50 to −15 °C, demonstrating the material's low-temperature stability. However, for sensing materials in polar environments, a broad damping temperature range was more critical. PUFT$_2$-50% IL exhibited good damping performance (tan $\delta > 0.5$) between −45 and −25 °C, which significantly increased its potential for

polar applications. Due to its observed damping behavior, PUFT$_2$-50% IL was also suitable for protecting fragile objects. For example, as shown in Fig. 4b, when a 100 g ball was dropped from a height of 50 cm, unprotected glass slides shattered completely. In contrast, a 0.7 mm thick PUFT$_2$-50% IL film effectively prevented glass breakage under both room temperature and cryogenic conditions (Supplementary Movie 1).

Furthermore, considering that PUFT$_2$-50% IL possessed a low $T_g$, this material was expected to exhibit temperature-induced shape memory behavior. The shape memory properties were quantitatively investigated using DMA (Fig. 4c). The results demonstrated that the shape recovery rate of PUFT$_2$-50% IL increased with cycling, rising from 86.5 to 94.8%, while its fixation rate remained stable at approximately 98.3%. This increase in recovery rate was likely attributed to the gradual alignment of molecular chains along a consistent orientation and reduced chain entanglements during cyclic loading. These quantitative results verified the shape memory capability of PUFT$_2$-50% IL. Here, the focus was primarily on the material's unidirectional shape memory

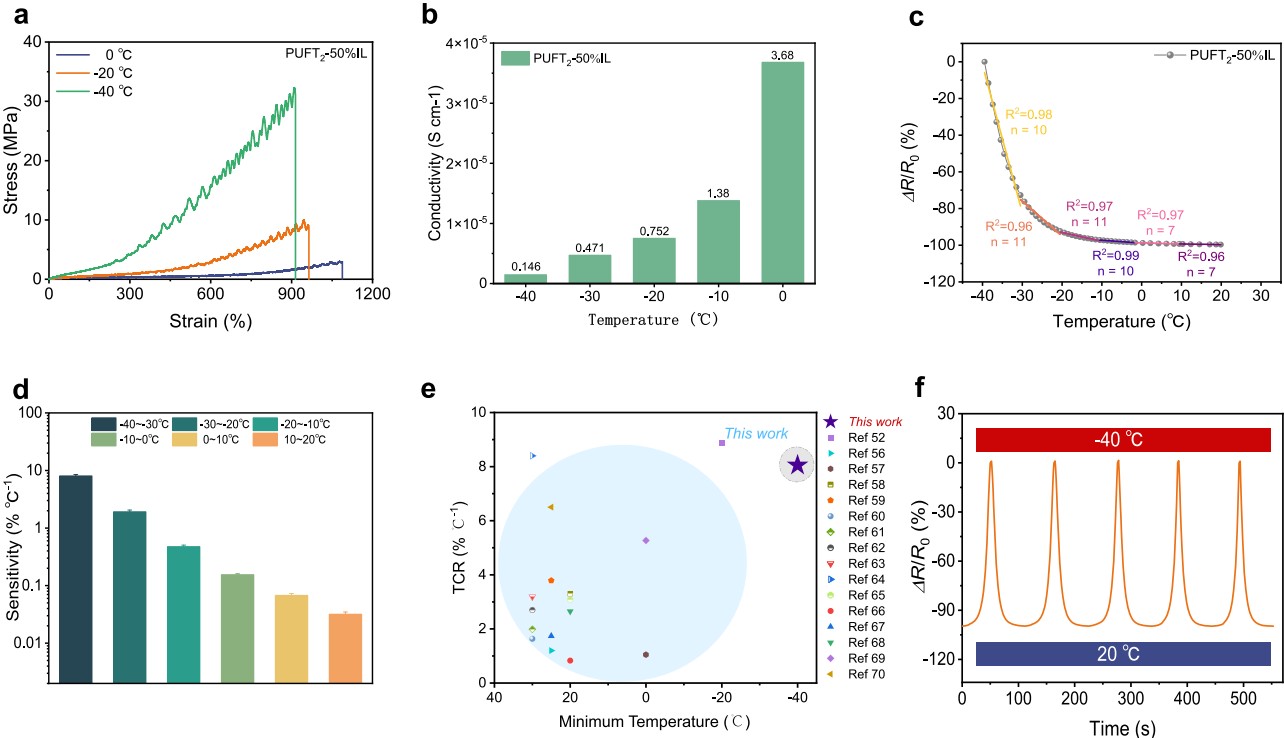

**Fig. 5 | Low-temperature tolerance and temperature sensitivity. a** Stress–strain curves of PUFT$_2$-50% IL at different temperatures. **b** Ionic conductivity of PUFT$_2$-50% IL at different temperatures. **c** The relative resistance change of PUFT$_2$-50% IL in the temperature range of −40 to 25 °C with the solid line representing the linear fits. **d** Temperature coefficient of resistance (TCR) of PUFT$_2$-50% IL. Data are presented as mean values ± SEM (n presented in Fig. 5c). **e** Comparison of TCR and minimum temperature of other reported samples. **f** The relative resistance changes in cyclic tests between −40 and 20 °C.

effect and its potential for smart wearable applications. Specifically, the ionically conductive elastomer could be coiled on flat surfaces or fingers and fixed into a temporary shape at low temperatures. This temporary shape then recovered to nearly indistinguishable original configurations within a short time at room temperature (Fig. 4d; Supplementary Movies 2 and 3). Notably, simulated finger-mounted demonstrations revealed that the material maintained its coiled state for temperature monitoring under extreme cryogenic conditions without restricting normal finger flexibility. Upon returning to room temperature, the material reverted to its original shape and conformed to the finger, enabling subsequent human motion detection. These capabilities highlighted the material's promising potential for intelligent wearable devices and signal transmission in polar environments.

### Low temperature tolerance and temperature sensitivity

Existing flexible sensors commonly suffer from instability under varying environmental conditions. As previously mentioned, PUFT$_2$-50% IL exhibited good damping performance, strongly suggesting its potential for cryogenic stability. Figure 5a demonstrated the typical stress–strain curves of PUFT$_2$-50% IL at 0, −20, and −40 °C. Notably, with continuous temperature reduction, the material's strength increased while its elongation at break decreased. Compared with room temperature conditions, the mechanical properties showed varying degrees of enhancement (Supplementary Fig. 15). Remarkably, at −40 °C, the strength and toughness reached 32.31 MPa and 107.05 MJ m$^{-3}$, representing 2348 and 1515% improvements, respectively, while still maintaining high stretchability (Supplementary Table 3). This phenomenon could be attributed to the restricted molecular chain mobility in polyurethane elastomers at lower temperatures, which intensified intermolecular interactions and facilitated the formation of ordered structures. Consequently, the polymer chains exhibited reduced sliding capacity and deformation capability,

thereby increasing the resistance to plastic deformation in the elastomer (the serrated stress–strain patterns observed were attributed to the liquid nitrogen cooling process employed during tensile testing). Similarly, ionic conductivity depends on ion mobility, which was intrinsically linked to polymer chain conformational changes and thermal motion. To investigate the electrical performance of PUFT$_2$-50% IL under cryogenic conditions, EIS measurements were conducted within the temperature range of −40 to 0 °C (Supplementary Fig. 16). As shown in Fig. 5b, the ionic conductivity of PUFT$_2$-50% IL demonstrated a progressive decline with decreasing temperature. This reduction stemmed from strengthened intermolecular interactions at lower temperatures, leading to denser polymer networks that restricted ion transport channels and impeded ionic migration. Notably, even at −40 °C, PUFT$_2$-50% IL maintained a relatively high ionic conductivity of $1.46 \times 10^{-6}$ S cm$^{-1}$, significantly expanding its operational temperature range as a functional sensor material.

Benefiting from its favorable combination of low-temperature toughness, ionic conductivity, and stretchability, PUFT$_2$-50% IL demonstrated promising potential as a resistive temperature sensor for cryogenic applications. Figure 5c illustrated the temperature-dependent relative resistance ($\Delta R/R_0$) curve of PUFT$_2$-50% IL. Notably, the relative resistance values at all tested temperatures were negative compared to the initial resistance at −40 °C, indicating a significant decrease in resistance with increasing temperature. This observation confirmed that PUFT$_2$-50% IL exhibited negative temperature coefficient behavior. To quantitatively evaluate its sensitivity and stability as a temperature sensor, we introduced the TCR, defined as the relative resistance change per unit temperature variation. The TCR value, derived from the slope of the linear fitting curve in Fig. 5c, reflected the sensor's sensitivity, with higher absolute TCR values corresponding to greater sensitivity. As shown in Fig. 5d, the absolute TCR of PUFT$_2$-50% IL reached a maximum of 8.05% °C$^{-1}$ in the −40 to −30 °C range. Remarkably, as shown in

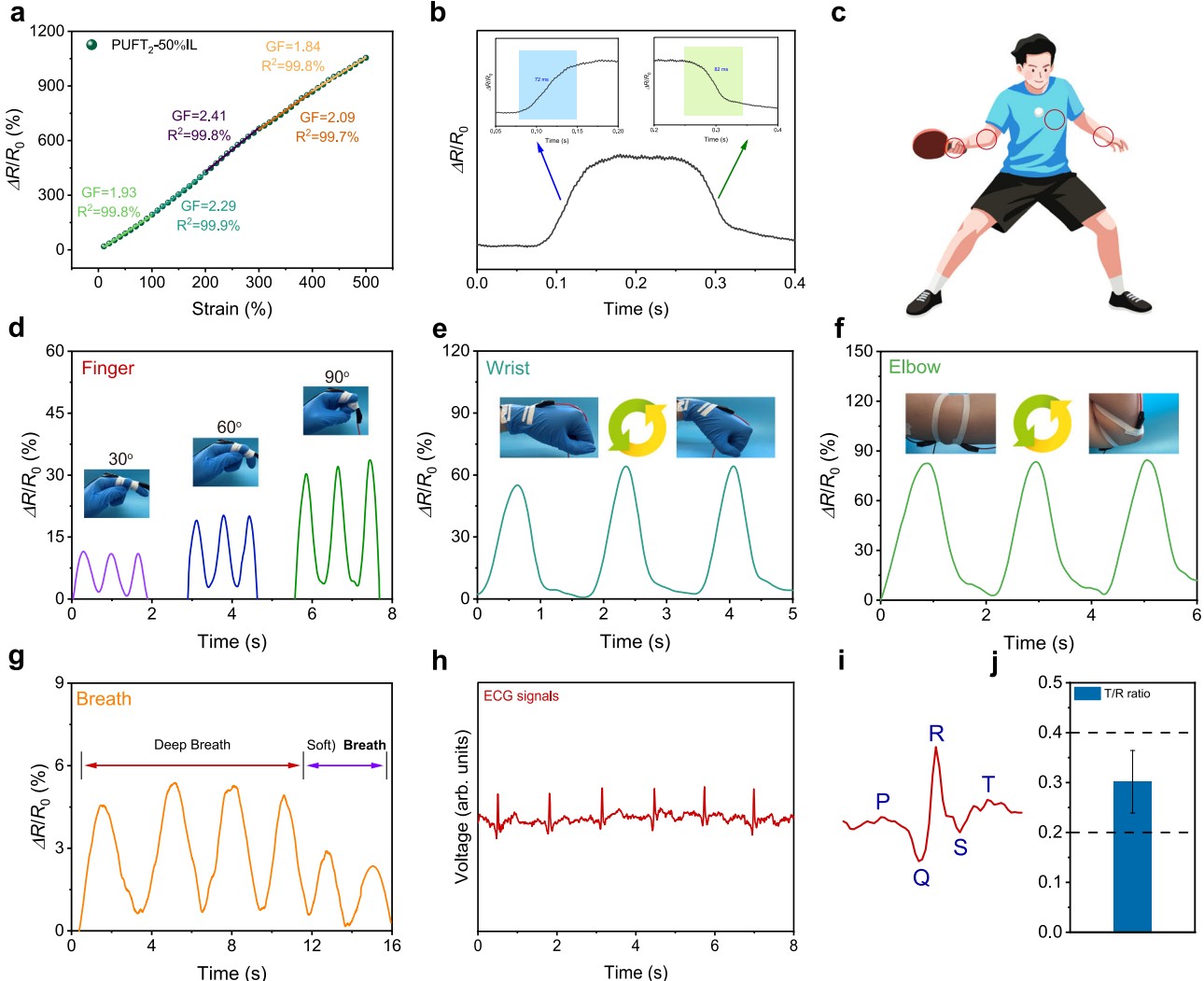

**Fig. 6 | Sensing performance and health monitoring applications. a** Strain sensitivity curve of PUFT$_2$-50% IL. **b** Response and recovery times. **c** Schematic diagram illustrating the placement of PUFT$_2$-50% IL in various parts of the human body. **d** $\Delta R/R_0$ of PUFT$_2$-50% IL during finger bending at different angles. **e** $\Delta R/R_0$ of PUFT$_2$-50% IL due to wrist flexion. **f** $\Delta R/R_0$ of PUFT$_2$-50% IL due to elbow flexion. **g** $\Delta R/R_0$ of PUFT$_2$-50% IL due to different breath patterns. **h** ECG signals detected with the PUFT2-50% IL electrode. **i** A period of ECG signal detected by PUFT$_2$-50% IL. P wave, QRS complex, and T wave are seen. **j** T/R ratio of ECG signals. Data are presented as mean values ± SD ($n$ = 3).

Fig. 5e, this material operated at a temperature of −40 °C while maintaining a high TCR, a performance that is notable among reported flexible temperature sensors[49,53–67]. Furthermore, the temperature-resistance relationship in viscoelastic ionic elastomers adhered to the Vogel–Tamman–Fulcher equation, which governed temperature-dependent ionic conductivity. The coefficient of determination ($R^2$) of the linear fitting curve determined whether temperature could be accurately represented by resistance variations. As annotated in Fig. 5d, all $R^2$ values exceeded 0.96 across tested temperature ranges, demonstrating the precise quantifiability of temperature through resistance changes. This combination of high TCR sensitivity and robust linear correlation underscored PUFT$_2$-50% IL's viability as a high-performance temperature-sensing material in extreme cold environments. Notably, the PUFT$_2$-50% IL sensor demonstrated good temperature-response cyclability, as evidenced by its stable relative resistance variation during five consecutive thermal cycling tests between −40 and 20 °C (Fig. 5f). Furthermore, its low-temperature stability was confirmed by a minimal resistance change of less than 4% over a continuous 120-min test at −40 °C (Supplementary Fig. 17), underscoring its strong potential for

reliable applications in cryogenic environments. These robust performances under periodic thermal shocks confirmed the material's structural integrity and temperature-tolerance capability, critical for reliable operation in extreme environments.

### Highly sensitive and healing ionic skin

Strain sensors represent an ideal direction for future development of flexible wearable sensors. As shown in Supplementary Fig. 18, the PUFT$_2$-50% IL demonstrated the capability to detect distinct signal variations induced by minor strain changes (20%) within a 20–100% strain range. Meanwhile, the sensitivity of PUFT$_2$-50% IL was evaluated under strain conditions ranging from 0 to 500% (Fig. 6a). The gauge factor was determined to be 1.93 within the 0–100% strain range and was observed to increase with increasing strain, reaching a value of 2.41 at 200–300% strain. Notably, the sensor exhibited rapid response characteristics with measured response and recovery times of 72 and 82 ms respectively, indicating high stability and sensitivity (Fig. 6b). To evaluate its potential as an epidermal biosensor, PUFT$_2$-50% IL was employed for real-time monitoring of diverse human movements and

physiological signals (Fig. 6c). Figure 6d demonstrated that periodic finger flexion-extension motions induced corresponding cyclic resistance variations, through which the sensor accurately identified different bending angles while maintaining stable signal output across multiple cycles, thereby confirming its reliability and sensitivity. Furthermore, as illustrated in Fig. 6e, f PUFT$_2$-50% IL effectively detected wrist and elbow flexion with consistent resistance signal outputs. These quantifiable electrical signals enabled precise recording and analysis of athletes' motion patterns during training sessions, facilitating timely evaluation of movement accuracy and subsequent improvement of training efficiency.

In addition to the strong resistance signals described above, PUFT$_2$-50% IL successfully detected subtle physiological activity signals. Respiratory monitoring, which serves as an indicator of exercise intensity or health status, constitutes an essential component of comprehensive health monitoring systems. As shown in Fig. 6g, thoracic expansion and contraction during breathing generated strain variations in the attached PUFT$_2$-50% IL, which induced measurable resistance changes. The sensor differentiated between deep and shallow breathing patterns with precision, further confirming its stability and sensitivity. Although respiratory rhythm reflects autonomic nervous system activity and its detection accuracy directly influences metabolic state assessments, the limitations of single-parameter respiratory monitoring necessitated the application of PUFT$_2$-50% IL as an electrode for ECG detection. Figure 6h illustrated the stable ECG signals acquired using PUFT$_2$-50% IL. Signal quality was evaluated through characteristic waveforms, including the P-wave, QRS complex, and T-wave, which correlated with cardiac electrical activity and were critical for clinical diagnosis (Fig. 6i). The T-wave amplitude, representing ventricular repolarization, typically ranges between 20 and 35% of the R-wave amplitude associated with ventricular depolarization. Clinically acceptable ECG signal quality requires a T/R peak amplitude ratio within 0.2–0.4[46,68]. As demonstrated in Fig. 6j, the T/R ratios derived from PUFT$_2$-50% IL-detected ECG signals consistently fell within this range, confirming reliable signal acquisition. These results collectively demonstrated the high sensitivity and stability of PUFT$_2$-50% IL in sensing applications.

Previously, the self-healing capability of PUFT$_2$-50% IL had already been confirmed. Therefore, it could be inferred that PUFT$_2$-50% IL would still be able to maintain its sensing ability after healing. Healed samples (1 h healing) successfully monitored repetitive finger and wrist flexion cycles, with $\Delta R/R_0$ amplitude consistency confirming stable real-time motion detection (Supplementary Figs. 19 and 20). Furthermore, the healed material maintained stable ECG signal acquisition capability post-repair, as evidenced by unaltered signal fidelity in Supplementary Fig. 21. These results collectively demonstrated that the self-healing mechanism preserved both mechanical integrity and sensing functionality, significantly extending the material's applicability in demanding environments. This work substantiated that hierarchical dynamic bonding networks not only enabled robust self-healing but also ensured performance retention across mechanical, electrical, and bio-signal sensing modalities.

## Discussion

In this work, we developed a fluorinated polyurethane elastomer with a synergistic dynamic network through rational molecular design, integrating environmental stability and damping capabilities. The elastomer exhibited high stretchability (>900%), ionic conductivity ($1.34 \times 10^{-4}$ S cm$^{-1}$), shape memory effect (shape recovery rate up to 94.8%) and self-healing properties (85.4% for 1 h), functioning as a strain sensor with high sensitivity and operational stability (response time of only 72 ms). It reliably detected both strong resistance signals from body movements across multiple joints and weak resistance signals associated with physiological activities, such as breath and ECG

monitoring. Notably, the material achieved clinically compliant signal fidelity in ECG detection, as evidenced by T-wave to R-wave amplitude ratios within the 0.2–0.4 range. Remarkably, the healed elastomer maintained accurate and stable sensing performance for health monitoring. Furthermore, the material demonstrated cryogenic stability, retaining robust mechanical properties (strength up to 32.31 MPa and toughness up to 107.05 MJ m$^{-3}$) and preserved ionic conductivity ($1.46 \times 10^{-6}$ S cm$^{-1}$) at −40 °C, highlighting its significant potential for practical applications in extreme environments. Additionally, the elastomer displayed a high TCR of 8.05% °C$^{-1}$ in the ultra-low temperature range of −40 to −30 °C, meeting the critical requirement for temperature-sensing functionality in polar regions.

## Methods

### Materials

Polycarbonate diol (PCDL, $M_n$ = 2000 Da, 99%) was purchased from Liduo Chemical Co., Ltd. (Jining, China). Isophorone diisocyanate (IPDI, 99%), MDH (97%) and dibutyltin dilaurate (DBTDL, 98%) were purchased from Energy Chemical Co., Ltd. (China). 2,2'-bis(trifluoromethyl) benzidine (TFMB, 98%), (2-hydroxyethyl) trimethylammonium bromide (TLB, 98%) and 1-ethyl-3-methylimidazolium bis(trifluoromethanesulfonyl) imide ([EMIM]$^+$ [TFSI]$^-$, 99 %) were purchased from TCI (Shanghai) Development Co., Ltd. $N$, $N$-dimethylformamide (DMF, 99.5%) was purchased from Rionlon Bohua Pharmaceutical and Chemical Co., Ltd. (Tianjin, China). All other chemicals and solvents were of analytical grade and used without further purification.

### Synthesis of the PUFT$_x$ elastomer

For the synthesis of the PUFT$_1$: PCDL ($M_n$ = 2000 Da, 5.0 mmol, 10.00 g) was mechanically stirred at 110 °C for 2 h under nitrogen (the following operations are carried out under nitrogen atmosphere if not otherwise specified.) in a three-necked round-bottom flask to remove residual moisture, cooled to 80 °C, and supplemented with a solution of IPDI (10.0 mmol, 2.25 g) and DBTDL (0.02 g) in DMF (40 mL). The mixture was stirred at 80 °C for 3 h, and the resulting prepolymer solution was supplemented with MDH (4.0 mmol, 0.53 g) and TFMB (0.5 mmol, 0.16 g). The mixture was heated at 80 °C for 2 h to a chain extension reaction, then supplemented with TLB (1 mmol, 0.18 g), further heated at 80 °C for 5 h, until the reaction was completed. The resulting liquid was poured into clean Teflon molds, and the samples were vacuum-dried at 80 °C for 48 h. The product was named PUFT$_1$. Following the same procedure for PUFT$_x$ were synthesized by changing the feed mass of MDH, TFMB and TLB. The mixing ratios used to prepare PUFT$_x$ are given in Supplementary Table 4.

### Synthesis of the PUFT$_x$-$y$IL elastomer

For the synthesis of the PUFT$_2$-50% IL: PUFT$_2$ (2.00 g) was dissolved in DMF (40 mL) and [EMIM]$^+$ [TFSI]$^-$ (2.00 g) was added after the dissolution was completed. After 1 h of stirring to mix well, the mixture solution was poured into clean Teflon molds, and the samples were vacuum-dried at 80 °C for 48 h. The product was named PUFT$_2$-50% IL.

### General characterizations

TGA was evaluated in an SDT 650 synchronous thermal analyzer (TA, America) in the range of 25–800 °C under a nitrogen flow (100 mL min$^{-1}$) at a heating rate of 10 °C min$^{-1}$. DSC measurements (DSC STA449F3, Netzsch, Germany) were performed in a flow of nitrogen. The samples (5–6 mg) were heated to 150 °C, cooled to −100 °C, and reheated to 150 °C at a rate of 10 °C min$^{-1}$. XRD measurements were characterized on a Bruker D8 Advance (BRUKER, Germany) using Cu-Kα radiation with $\lambda$ = 1.5418 Å, and the data were recorded in a range from 5° to 90° at a scanning speed of 5° min$^{-1}$. FT-IR spectra was recorded on a Nicolet iS20 (Thermo Scientific, America)

spectrometer from 500 to 4000 cm$^{-1}$ in attenuated total reflectance mode. Hydrogen proton nuclear magnetic resonance ($^1$H NMR) spectra were measured on an AVANCE NEO 400 MHz spectrometer (Bruker, Germany), using CDCl$_3$ as solvent. Raman spectroscopy was recorded using a Raman spectrometer (LabRAM HR Evolution, HORIBA), with the laser power of 25 mW for 532 nm laser excitation. XPS was carried out on K-Alpha (Thermo Scientific, America), excited by an Al Kα (hv = 1486.6 eV) Mono X-ray with a beam spot of 400 μm. ToF-SIMS measurements were conducted utilizing a PHI nanoTOF II Time-of-Flight SIMS (ULVAC-PHI. INC, JAPAN) instrument. Ion mass spectrometry images were acquired by sputtering the sample surface using a 30 keV Bi$_3^{++}$ ion beam in high mass resolution mode over a surface area of 200 × 200 μm of the test sample. The microphase structure of the samples was detected by an atomic force microscope (AFM, Bruker Dimension Icon, Germany) in tapping mode with a scan area of 0.6 × 0.6 μm. Thermomechanical properties were characterized using a dynamic mechanical analyzer (TA Q850 DMA). To that end, all samples were cut into rectangles (20 × 4 × (0.2–0.5) mm) and heated from −120 to 100 °C with a heating rate of 5 °C min$^{-1}$ and a frequency of 1 Hz.

## Data availability

The authors declare that data supporting the findings of this study are available within the paper and its Supplementary Information Files. All data are available from the corresponding author upon request. The modeling details for quantum chemical calculations and molecular dynamics simulations have been uploaded in figshare (https://doi.org/10.6084/m9.figshare.30484481). Data generated in this study are provided in the Source data file with this paper. Source data are provided with this paper.

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

## Acknowledgements

This work was financially supported by the National Natural Science Foundation of China (Grant No. 52375214, X.Z.), the CAS Project for Young Scientists in Basic Research (YSBR-023, X.Z.), the Youth Innovation Promotion Association of the Chinese Academy of Sciences (Y2022103, X.Z.), the Major Program of the Lanzhou Institute of Chemical Physics, CAS (No. XDB0470303, Q.W.), and the Excellent Doctoral Project of the Province Natural Science Foundation of Gansu (24JRRA071, J.X.).

## Author contributions

T.C., J.X. and X.Z. provided experimental ideas, wrote the manuscript and materials synthesis. C.W. and X.P. participated in the design of figures. T.W. and Q.W. revised the first draft.

## Competing interests

The authors declare no competing interests.
