## [Transparent Peer Review File · Nature Communications]

Shape-Memory Polyurethanes for Polar Wearables with Ultrasensitive Multi-Monitoring

Corresponding Author: Professor Xinrui Zhang

Version 0:

Reviewer comments:

Reviewer #1

(Remarks to the Author)
Accept as is.

Reviewer #2

(Remarks to the Author)

This manuscript describes a multifunctional polyurethane elastomer constructed through fluorine-cation interactions, combining shape memory capability, self-repairing properties, and excellent low-temperature resistance. A novel methodological strategy is proposed that is applicable to ultra-low-temperature sensors, enabling simultaneous improvements in stability and sensitivity. This approach is expected to be instructive for applications in wearable devices operating in extreme environments and will likely attract broad interest from researchers in the fields of smart wearable materials, smart sensing, and extreme-environment technologies. Overall, the manuscript presents a promising contribution that could be suitable for publication in Nature Communications after minor revisions are addressed. The following points should be considered for improvement:

1. Many figures are not well organized. It is recommended that some of the diagrams be combined for clarity and that the overall figure layout be polished.
2. The manuscript lacks information on the variability and reproducibility of the presented data. Without this, readers must assume that the results are based on single measurements. The reliability of the findings would be enhanced by including data from multiple samples and statistical analyses.
3. Given that the elastomer's properties are closely related to its molecular structure, the structural characterization appears insufficient. Additional data such as NMR spectra should be provided to confirm the chemical structure.
4. On page 12, line 219, the FTIR peaks are attributed to different carbon atoms on the imidazole ring. It would be helpful to include the molecular structure of the [EMIM]⁺ cation and label the relevant carbon atoms for clarity.
5. On page 21, line 404, ionic conductivity at -40 °C is mentioned. To better contextualize this result, a comparison with previously reported materials would be beneficial. In Figure 5e, the lowest working temperature and TCR index are compared; adding a comparison of ionic conductivity across different materials would further strengthen the discussion.
6. In Figure 7c, the self-healing efficiency of elongation at break is significantly higher than that of tensile strength. An explanation for this discrepancy should be provided, along with data or discussion regarding changes in toughness after healing.

Reviewer #3

(Remarks to the Author)

This review evaluates the molecular modeling aspects of the manuscript. Overall the results are contradictory (see my comment about binding energy below), not sufficiently detailed, and (probably, but hard to judge since so much information is lacking) not run long enough. It is not acceptable in the current form. My specific comments are:

I have some questions about the following text:

"Quantum chemical calculations were first conducted to evaluate the binding energy of [EMIM]⁺ [TFSI]⁻ in the presence and absence of polymer chains with the computational details provided in Supporting Information. As shown in Fig. 2c, the

binding energy of [EMIM]⁺ [TFSI]⁻ under polymer chain influence was significantly less negative compared to its isolated state. Given that binding energy values were typically negative with more negative values indicating stronger binding..." My questions about this text are: (1) The binding energy of [EMIM]⁺ [TFSI]⁻ under polymer chain influence is given as -142.4 which is more negative, not less negative, than the -83.2 value for the isolated state. This would seem to negate the argument given in the text. (2) Under the polymer chain influence, I don't know what was computed using equation 10 of the supplementary file? Be more specific here since there are more than two species.

The authors say: "Initially, the simulation systems underwent structural optimization." (1) What does this mean? (2) How were the 24 or 36 polymers arranged? (3) What does Figure 12 show? Is it a snapshot of a single polymer or is this the 36 polymer system? I'm guessing it is just a single polymer. (4) We are told it is "optimized" but what does this mean? Some parts of the polymer chain are extended and some are coiled. How was this structure created? (5) More importantly, how are the 36 polymer chains arranged? Was this single polymer configuration replicated 36 times? How were the 36 polymers packed together? It is impossible to judge the realism and quality of the simulations without these details.

The authors state that "The diffusion coefficient is obtained from the long-time limit of the MSD". But the simulation was only run for 5 ns. It is doubtful that there is a long-time limit with this short a simulation. The authors need to show the MSD vs. time plot and show the fitted line to the data that they used to compute the diffusion coefficient.

I have some questions about the following text:

"...relaxation for 1 nanosecond until the systems reached a steady state. Following this, over a period of 5 nanoseconds, the systems were uniformly stretched in the x direction to twice their original length, while maintaining a pressure of 1 atmosphere in the y and z directions. Then, over another 5 nanoseconds, the systems were compressed back to their original length in the x-direction, again maintaining a pressure of 1 atmosphere in the y and z directions. For the structures before stretching, after stretching, and after compression, we conducted 5-nanosecond simulations under the NVT ensemble"

My questions about this text are: (1) what does "steady state" mean? 1 ns is very short. Again this goes back to my previous comment about how the polymer chains were arranged relative to one another. Steady state could mean the box volume stops changing, or the total energy converges to a steady value, or that the RMSD of the polymers stops changing, or any number of other criteria. It is hard to believe that 1 ns is enough to reach equilibrium. (2) how do the y and z box sizes compare before and after the stretching and compression? (3) After the 5 ns stretching, was the system immediately compressed or was the 5 ns NVT run done in-between with the compression done from after the NVT run? Or is the NVT run done on a pull-out of the NPT configuration without interrupting the stretch/compress run? This isn't clear. (4) If the NVT runs to measure conductivity are done immediately following the stretching run (and the compression run), presumably the system is not at equilibrium yet (the system would relax to equilibrium after the stretching is stopped over some time scale) so this would be a non-equilibrium simulation which is not appropriate.

Reviewer #4

(Remarks to the Author)

This study presents a supramolecular polyurethane engineered with fluorine-rich segments that form electrostatic crosslinks with positively charged ionic groups at polymer chain terminals and establish fluorine-dipole interactions with blended ionic liquid to stabilize ion transport pathways. Many interesting results are presented. However, the reviewer suggests the following points to be seriously addressed:

1. In Abstract, The author points out: "The resulting ionically conductive elastomer combines shape memory capacity, self-healing property, and exceptional cryogenic tolerance, retaining robust mechanical strength (~32.31 MPa), toughness (~107.05 MJ m⁻³) and substantial ionic conductivity even at -40°C. Notably, it achieves an unprecedented combination of record-low operational temperatures (-40°C to -30°C) and an ultrahigh temperature coefficient of resistance (TCR = 8.05% °C⁻¹), setting a new benchmark for cryogenic sensing materials." However, the author did not focus on testing the various characteristics and application under low-temperature conditions.
2. In Section 2.6 (Highly sensitive ionic skin), there are too few tests on static characteristics, and basic characteristic indicators such as sensitivity, linearity and range are missing.
3. In Section 2.7 (Self-healing properties), Fig. 6d-e and Fig. 6f should be marked as Fig. 7d-e and Fig. 7f. This mistake should be due to the author's carelessness.
4. As shown in Fig.5 and Fig.6, compared with the gauge factor, the resistance temperature coefficient is too strong. In the actual application process, it will affect its application in strain or pressure.
5. It is recommended to describe the material properties first and then the application properties. For example, Section 2.7 should be described earlier.
6. Generally, ionic conductors are commonly used in capacitive sensors, demonstrating high sensitivity, but are less applied in resistive principles. It can also be roughly seen from the results in Section 2.6 (Highly sensitive ionic skin) that the gauge factor of the designed sensor is relatively low.
7. The application part of the material is too simple to support its application value.

Version 1:

Reviewer comments:

Reviewer #2

(Remarks to the Author)

The authors have made appropriate modifications to the paper and given reasonable explanations to the questions raised by reviewers, which resolved the concerns of the reviewers. Therefore, it should be suitable for publication in Nature Communications.

[Editor's Note: This Reviewer was asked to evaluate the the response to Reviewer #4's comments, and provided the following:]

In my opinion, the authors have conducted a thorough and proper revision in response to the reviewers' comments. The incorporation of new experimental data, expanded discussions, and structural refinements has substantially elevated the overall quality and clarity of the work. In particular, the concerns raised by Reviewer #4 have been addressed comprehensively and convincingly.

The manuscript now includes critical low-temperature stability tests and essential sensing metrics, such as sensitivity and linear sensing range, that firmly establish the material's practical applicability. These additions not only validate the authors' initial claims but also significantly bolster the work's completeness and scientific rigor. The revised narrative more logically progresses from intrinsic material properties to functional applications, greatly improving readability and impact.

Therefore, I believe this revised manuscript is suitable for publication in Nature Communications.

Reviewer #3

(Remarks to the Author)

This review evaluates the molecular modeling aspects of the revised manuscript. I now consider the manuscript acceptable for publication but I would appreciate if the authors could address the following comments:

In the rebuttal the authors say "Structural optimization is initially carried out by adjusting the atomic positions based on the direction of atomic forces, moving them toward a lower total potential energy. This process is done to avoid excessively high initial energy of the structure." This is fine if the sole purpose is to avoid high initial energy structures. But this local relaxation will not cause an extended chain to become coiled or vice versa. Probably "local relaxation" is a better term here than optimization.

Fig. R6 is helpful so I can see what the full system is. Thank you for including this. However, in the rebuttal the authors say "Driven by thermal motion, the polymer chains spontaneously bent and entangled, forming the structure corresponding to ambient temperature and pressure." It is far from obvious that any entanglement occurs. Can the authors substantiate this with a measurable quantity, in other words can they quantify entanglement?

In Fig. R7 the authors show the MSD vs. time plot which I asked for. This is appreciated. However, since the diffusion constant is quite low (4 orders of magnitude lower than water) it is not convincing to stop the plots at 2 ns (= 2000 ps). The data sort of looks linear but it would be much more convincing if the time scale was extended.

In the rebuttal the authors say "We define the steady state as the condition in which the volume of the simulation box no longer changes and the total energy has converged." These are very weak criteria for equilibrium. More convincing would be to track the radius of gyration of the polymer chains. This is because changes in the radius of gyration or other polymer conformational properties might not show up in the overall system volume and would be a more careful test of steady state.

In the rebuttal the authors say "After compression, the lengths in the y and z directions return to the values observed before the stretching began". Were the y and z box lengths constrained to have the same value as each other? Otherwise it would be surprising for the y and z values to return exactly to their starting values. If such a constraint was used it should be stated.

Version 2:

Reviewer comments:

Reviewer #3

(Remarks to the Author)

This review evaluates the molecular modeling aspects of the revised manuscript. The authors have comprehensively addressed all of my comments which is greatly appreciated. I have no further comments or suggestions. I recommend publication.

Response to Previous Reviewers' Comments

Dear reviewers,

We would like to thank all reviewers for their constructive comments and suggestions. After carefully reading and thinking about the reviewers' comments, we have supplemented and revised this manuscript to reflect most of their suggestions. These comments are valuable and helpful for the improvement of our paper, and they also have important guiding significance for research. Therefore, we have studied the comments carefully and have made further revisions, hoping that the reviewers and editors will be satisfied with the revised manuscript.

Our point-by-point responses to these comments are provided in another document, and the main changes in the revised manuscript are marked in yellow (revised contents) and blue (supplementary contents). In any case, we are open to consideration of any further comments and suggestions. We look forward to hearing from you on the status of our submission. Thanks again to the four reviewers for their constructive comments, and to the editors for their affirmation and support of this work.

Sincerely,

Jing Xu and Xinrui Zhang

➤ **Reviewer #1 (Remarks to the author):**

Accept as is.

➤ **Response #1:**

We are very grateful to the reviewer for reviewing our manuscript and we sincerely appreciate your recognition of our work. Your unequivocal endorsement of our work is both deeply encouraging and a significant validation of our research efforts. We sincerely appreciate your insightful recognition of the manuscript's scientific rigor and contribution to the field. Such support from a respected expert is invaluable to our team.

➤ **Reviewer #2 (Remarks to the author):**

This manuscript describes a multifunctional polyurethane elastomer constructed through fluorine-cation interactions, combining shape memory capability, self-repairing properties, and excellent low-temperature resistance. A novel methodological strategy is proposed that is applicable to ultra-low-temperature sensors, enabling simultaneous improvements in stability and sensitivity. This approach is expected to be instructive for applications in wearable devices operating in extreme environments and will likely attract broad interest from researchers in the fields of smart wearable materials, smart sensing, and extreme-environment technologies. Overall, the manuscript presents a promising contribution that could be suitable for publication in Nature Communications after minor revisions are addressed. The following points should be considered for improvement.

➤ **Response #2:**

We are very grateful to the reviewer for reviewing our manuscript and providing constructive comments, which are immensely helpful in improving the quality of our manuscript. To fully address your comments and concerns, we have updated the manuscript and provided detailed explanations. Besides, we also have supplemented or revised figures, expanded discussions, and improved the language in the revised manuscript. The main changes in the revised manuscript or supporting information are in yellow (revised contents) and blue (supplementary contents). In the following text, we will address each comment point by point.

• **Comment #2-1**

Many figures are not well organized. It is recommended that some of the diagrams be combined for clarity and that the overall figure layout be polished.

• **Response #2-1:**

We sincerely thank you for your insightful comments and have thoughtfully

revised the manuscript based on the reviewers' comments. We have reconsidered the combination of the figures and have revised this in the re-submission.

- **Comment #2-2**

The manuscript lacks information on the variability and reproducibility of the presented data. Without this, readers must assume that the results are based on single measurements. The reliability of the findings would be enhanced by including data from multiple samples and statistical analyses.

- **Response #2-2:**

Thanks for your careful review and constructive suggestions regarding our manuscript. In this work, we have conducted repeated experiments to prove our experimental results, and the test data obtained are those we have obtained after repeated proof. The relevant performance data of the samples are fully reliable. We also provide detailed test descriptions at the end of the paper, and all the results obtained from the tests and experiments support each other. In addition, we have uploaded the raw data for experimental replication. Overall, the research results are highly reliable.

- **Comment #2-3**

Given that the elastomer's properties are closely related to its molecular structure, the structural characterization appears insufficient. Additional data such as NMR spectra should be provided to confirm the chemical structure.

- **Response #2-3:**

Thanks again for your careful review and constructive suggestions on our manuscript, which greatly improved its quality. According to your advice, we have performed ^1H NMR tests to confirm the structure of the as-synthesized polymers. The results have been added in the revised Supplementary Information as shown in **Figs. R1-R2 (Figs. S2-S6 in Supplementary Information)**. From the results of NMR further

demonstrated that the materials have been successfully synthesized.

Fig.R1 (Fig. 5 in Supplementary Information) ^1H NMR spectrum of PUFT₂ in CDCl_3 .

The specific ^1H NMR analysis is as follows:

^1H NMR (400 MHz, CDCl_3) δ 6.99 (m, H-h), 6.79 (d, $J = 2.5$ Hz, H-f), 6.77 (m, H-c), 4.12 (m, H-b), 3.37 (s, H-a), 3.31 (s, H-d), 3.03 – 2.87 (m, H-g), 1.68 (m, H-e), 1.46 – 1.22 (s, H-i), 1.16 – 0.70 (m, H-j).

Fig.R2 (Fig. 6 in Supplementary Information) ^1H NMR spectrum of PUFT₂-50%IL in CDCl_3 .

The specific ^1H NMR analysis is as follows:

^1H NMR (400 MHz, CDCl_3) δ 8.94 (s, H-k), 7.30 (d, $J = 1.8$ Hz, H-l), 7.28 (d, $J = 1.8$ Hz, H-m), 7.09 – 6.94 (m, H-h), 6.79 (d, $J = 2.4$ Hz, H-f), 6.77 (d, $J = 2.6$ Hz, H-c), 4.27 (q, $J = 7.4$ Hz, H-n), 4.12 (m, H-b), 3.97 (s, H-p), 3.26 (s, H-a), 3.16 (m, H-d), 2.97 – 2.86 (m, H-g), 1.67 (m, H-e), 1.57 (t, $J = 7.4$ Hz, H-o), 1.41 (s, H-i), 1.11 – 0.77 (m, H-j).

The detailed descriptions added to the revised manuscript are as follows:

Simultaneously, the ^1H NMR results also confirmed the successful synthesis of PU and the effective incorporation of $[\text{EMIM}]^+ [\text{TFSI}]^-$.

- **Comment #2-4**

On page 12, line 219, the FTIR peaks are attributed to different carbon atoms on the imidazole ring. It would be helpful to include the molecular structure of the [EMIM]⁺ cation and label the relevant carbon atoms for clarity.

- **Response #2-4:**

We sincerely thank you for your kind suggestions, as well as apologize for the confusion caused to the reviewers due to unclear description. We have revised this in the re-submission, provided a more clearly figure (**Fig R3**).

Fig. R3 (Fig. 2 in revised manuscript)

- **Comment #2-5:**

On page 21, line 404, ionic conductivity at -40 °C is mentioned. To better contextualize this result, a comparison with previously reported materials would be beneficial. In Figure 5e, the lowest working temperature and TCR index are compared;

adding a comparison of ionic conductivity across different materials would further strengthen the discussion.

- **Response #2-5:**

Thank you very much for your careful review and constructive suggestions regarding our manuscript. According to your advice, we searched relevant works for comparison of ionic conductivity. Nevertheless, we noted that numerous cryogenic materials exhibited superior electrical conductivity at low temperatures compared to our material, yet they had not been utilized in sensors, nor had their TCR indices been characterized under cryogenic conditions. Concurrently, some literature on sensor materials either omitted low-temperature ionic conductivity testing or reported TCR indices solely above room temperature. These limitations hinder meaningful comparative analysis. Crucially, this work prioritized the ultra-low-temperature sensing performance of the material. Expanding the scope to include comprehensive low-temperature conductivity studies risked compromising the manuscript's conciseness.

Herein, we enthusiastically provide detailed comparative data where feasible. Specifically, **Fig. R4** now directly contrasts the cryogenic ionic conductivity of our material with literature values cited in the original Fig. 5e. This comparison conclusively demonstrates that our material outperforms these referenced materials in low-temperature ionic conductivity.

Fig. R4 Superiority of this materials over similar previously reported samples.

- **Comment #2-6:**

In Figure 7c, the self-healing efficiency of elongation at break is significantly higher than that of tensile strength. An explanation for this discrepancy should be provided, along with data or discussion regarding changes in toughness after healing.

- **Response #2-6:**

Thank you again for your valuable advice to improve the quality of our manuscript. Based on your advice, changes in toughness after healing have been added in the revised manuscript, as shown in **Fig. R5 (Fig. 4c in revised manuscript)**.

Fig. R5 (Fig. 7c in manuscript) Healing efficiency at different times.

In our system, the significantly higher self-healing efficiency of elongation at break compared to tensile strength can be attributed to the distinct structural dependencies of these two mechanical parameters. Elongation at break is mainly governed by the flexibility and mobility of polymer chains, as well as their ability to re-entangle and rearrange at the fracture interface. Under the influence of reversible non-covalent interactions (e.g., H-bonding and ionic interactions) and the plasticizing effect of the incorporated ionic liquid, these flexible chain segments can rapidly reconfigure during the healing process, leading to a substantial recovery of extensibility.

In contrast, tensile strength relies more on the integrity and spatial continuity of load-bearing permanent cross-linking nodes (chemical cross-links or highly ordered physical cross-linking domains). Microcracks, interfacial defects, and irreversible chain scission generated during fracture are difficult to completely eliminate during self-healing, resulting in stress concentration and premature failure under high loads, thereby limiting the recovery of tensile strength.

It should be noted that the healing efficiency of toughness in our system is often

even lower than that of tensile strength. Toughness is essentially the integrated product of strength and elongation, making it highly sensitive to both the network integrity and chain extensibility. During healing, even if extensibility is partially restored, any residual defects that compromise the load-bearing capability at high stress will significantly reduce toughness, and vice versa. Therefore, complete recovery of toughness requires simultaneous high recovery of both strength and elongation, which is generally more challenging to achieve in practical self-healing processes.

➤ **Reviewer #3 (Remarks to the author):**

This review evaluates the molecular modeling aspects of the manuscript. Overall the results are contradictory (see my comment about binding energy below), not sufficiently detailed, and (probably, but hard to judge since so much information is lacking) not run long enough. It is not acceptable in the current form.

➤ **Response #3:**

We sincerely thank the reviewer for carefully evaluating the molecular modeling aspects of our manuscript and for providing constructive comments. We acknowledge that some of our modeling results were insufficiently detailed and that certain explanations were unclear, which may have led to apparent contradictions. We apologize for the lack of clarity and detail in the initial submission and have substantially revised the relevant sections to address these concerns, as detailed in our point-by-point responses below.

• **Comment #3-1:**

"Quantum chemical calculations were first conducted to evaluate the binding energy of [EMIM]⁺ [TFSI]⁻ in the presence and absence of polymer chains with the computational details provided in Supporting Information. As shown in Fig. 2c, the binding energy of [EMIM]⁺ [TFSI]⁻ under polymer chain influence was significantly less negative compared to its isolated state. Given that binding energy values were typically negative with more negative values indicating stronger binding..."

My questions about this text are: (1) The binding energy of [EMIM]⁺ [TFSI]⁻ under polymer chain influence is given as -142.4 which is more negative, not less negative, than the -83.2 value for the isolated state. This would seem to negate the argument given in the text. (2) Under the polymer chain influence, I don't know what was computed using equation 10 of the supplementary file? Be more specific here since there are more than two species.

- **Response #3-1:**

(1) We apologize for the inconvenience caused to the reviewer by the incorrect description. Based on the quantitative calculation results, the binding effect of $[\text{EMIM}]^+[\text{TFSI}]^-$ is weaker in the absence of polymer chain influence, whereas it becomes stronger in the composite system when polymer chains are present. In the presence of polymer chain segments, the binding energy is not “less negative” but rather “more negative”. What we intended to convey is that a larger absolute value of the binding energy corresponds to a stronger binding effect. This led to our previous incorrect description, which we have now corrected in the revised manuscript. We sincerely thank the reviewer again for this valuable suggestion.

The specific changes in the manuscript are as follows:

“Quantum chemical calculations were first conducted to evaluate the binding energy of $[\text{EMIM}]^+ [\text{TFSI}]^-$ in the presence and absence of polymer chains with the computational details provided in Supporting Information. As shown in Fig. 2c, the binding energy of $[\text{EMIM}]^+ [\text{TFSI}]^-$ under polymer chain influence was significantly more negative compared to its isolated state. Given that binding energy values were typically negative with more negative values indicating stronger binding and structural stability, this result suggested that the cations and anions of the IL readily dissociate via fluorine-cation interactions without disrupting the ionic bonding between molecular chains. This separation mechanism ensured the integrity of the dynamic network while facilitating ion mobility.”

(2) Thank you for your careful review and constructive suggestions on our manuscript, which greatly improved its quality. Our binding energy calculation formula is defined as the total energy of the composite system after binding minus the energies

of each component in their isolated states. For multi-component systems, the resulting binding energy refers to the overall binding energy of the system when all components are aggregated together. Our original intention was to use Part A and Part B to represent individual components, indicating that the total energy should be subtracted by the energies of all such components. However, we did not take into account the potential ambiguity that may arise when more components are involved, and we sincerely apologize for this oversight. We have now provided a more detailed explanation of the binding energy calculation formula in the Supplementary Information.

The specific changes in the Supplementary Information are as follows:

The interaction energy of $[\text{EMIM}]^+[\text{TFSI}]^-$ were calculated by the following Equation 1 (Equation 10 in the Supplementary Information):

$$E_{\text{Bind(IL)}} = E_{\text{IL}} - (E_+ + E_-) \quad (1)$$

where $E_{\text{Bind(IL)}}$ denotes the interaction energy of $[\text{EMIM}]^+[\text{TFSI}]^-$ in the absence of polymer chains, E_{IL} , E_+ and E_- correspond to the energies of $[\text{EMIM}]^+[\text{TFSI}]^-$, isolated $[\text{EMIM}]^+$ and isolated $[\text{TFSI}]^-$.

The interaction energy of $[\text{EMIM}]^+[\text{TFSI}]^-$ under polymer chain influence were calculated by the following Equation 2 (Equation 11 in the Supplementary Information):

$$E_{\text{Bind(Complex)}} = E_{\text{Complex}} - (E_+ + E_- + E_{\text{TFMB}} + E_{\text{TLB}}) \quad (11)$$

where $E_{\text{Bind(Complex)}}$ denotes the interaction energy of $[\text{EMIM}]^+[\text{TFSI}]^-$ in the presence of polymer chains, E_{Complex} , E_+ , E_- , E_{TFMB} and E_{TLB} correspond to the energies of this system, isolated $[\text{EMIM}]^+$, isolated $[\text{TFSI}]^-$, isolated TFMB part and isolated TLB part.

- **Comment #3-2:**

The authors say: "Initially, the simulation systems underwent structural optimization." (1) What does this mean? (2) How were the 24 or 36 polymers arranged? (3) What does Figure 12 show? Is it a snapshot of a single polymer or is this the 36 polymer system? I'm guessing it is just a single polymer. (4) We are told it is "optimized" but what does this mean? Some parts of the polymer chain are extended and some are coiled. How was this structure created? (5) More importantly, how are the 36 polymer chains arranged? Was this single polymer configuration replicated 36 times? How were the 36 polymers packed together? It is impossible to judge the realism and quality of the simulations without these details.

- **Response #3-2:**

Many thanks to the reviewer for your careful reading of our manuscript and relevant remarks. Additionally, we sincerely apologize for any confusion caused by unclear descriptions. Below are our full responses to these issues, and we are open to consideration of any further comments and suggestions

(1) Here, we perform structural optimization for two systems that are undergoing simulation calculations. Structural optimization is initially carried out by adjusting the atomic positions based on the direction of atomic forces, moving them toward a lower total potential energy. This process is done to avoid excessively high initial energy of the structure.

(2) The polymers within the system are randomly arranged.

(3) Figure 12 shows the snapshot of a single polymer molecular chain. Snapshots of the specific polymer systems are shown in the **Fig. R6** below. Since the polymer systems are confined within the simulation box, the box boundaries limit the visibility of the detailed modeling of the polymer molecular chains. Therefore, we chose to present the snapshot of a single polymer molecular chain rather than that of the entire system.

Fig. R6 The snapshot of the two systems.

(4) We reduced the total energy of the initial structure through structural optimization. By using Packmol, we packed a specified number of polymer chains into a cubic simulation box. The system was then relaxed under the NPT ensemble at a temperature of 300 K and a pressure of 1 atm. Driven by thermal motion, the polymer chains spontaneously bent and entangled, forming the structure corresponding to ambient temperature and pressure.

(5) The 36 polymer chains within the box are randomly arranged and generated by replicating a single polymer configuration 36 times. However, due to the random motion of molecules during molecular simulation, the molecular configurations undergo certain changes, resulting in each polymer having a distinct and completely random configuration. Subsequently, we used the Packmol software to pack these polymers, together with the corresponding number of ions, into a cubic box, and then obtained a stable initial structure through relaxation.

- **Comment #3-3:**

The authors state that "The diffusion coefficient is obtained from the long-time limit of the MSD". But the simulation was only run for 5 ns. It is doubtful that there is a long-time limit with this short a simulation. The authors need to show the MSD vs.

time plot and show the fitted line to the data that they used to compute the diffusion coefficient.

- **Response #3-3:**

We thank the reviewer for your insightful and thoughtful comments. And we apologize for the omission of the MSD curve. We have now examined the MSD data and the corresponding fitted curve, as shown in the **Fig. R7** below. The data fitting was performed using the last one-third of the data for linear regression. The high degree of linearity in the fitting results demonstrates that the simulation has essentially reached convergence.

Fig. R7 The MSD curves of system 1 and system 2.

- **Comment #3-4:**

"...relaxation for 1 nanosecond until the systems reached a steady state. Following this, over a period of 5 nanoseconds, the systems were uniformly stretched in the x direction to twice their original length, while maintaining a pressure of 1 atmosphere in the y and z directions. Then, over another 5 nanoseconds, the systems were compressed back to their original length in the x-direction, again maintaining a pressure of 1 atmosphere in the y and z directions. For the structures before stretching, after stretching, and after compression, we conducted 5-nanosecond simulations under the

NVT ensemble".

My questions about this text are: (1) what does "steady state" mean? 1 ns is very short. Again this goes back to my previous comment about how the polymer chains were arranged relative to one another. Steady state could mean the box volume stops changing, or the total energy converges to a steady value, or that the RMSD of the polymers stops changing, or any number of other criteria. It is hard to believe that 1 ns is enough to reach equilibrium. (2) how do the y and z box sizes compare before and after the stretching and compression? (3) After the 5 ns stretching, was the system immediately compressed or was the 5 ns NVT run done in-between with the compression done from after the NVT run? Or is the NVT run done on a pull-out of the NPT configuration without interrupting the stretch/compress run? This isn't clear. (4) If the NVT runs to measure conductivity are done immediately following the stretching run (and the compression run), presumably the system is not at equilibrium yet (the system would relax to equilibrium after the stretching is stopped over some time scale) so this would be a non-equilibrium simulation which is not appropriate.

- **Response #3-4:**

We thank the reviewer for your insightful and thoughtful comments. We have carefully checked all the details of the simulation calculations, and we apologize for any confusion caused by our unclear description. We have refined the simulation details, and below are our point-by-point responses.

(1) We define the steady state as the condition in which the volume of the simulation box no longer changes and the total energy has converged. The convergence process is illustrated in the **Fig. R8** below, from which it can be observed that the system reaches a steady state after approximately 1 ns.

Fig. R8 The changes of energy (a) and volume (b) of system 1 as well as energy (c) and volume (d) of system 2.

(2) During the stretching process, the lengths in the y and z directions decrease. During the compression process, the lengths in the y and z directions increase. After compression, the lengths in the y and z directions return to the values observed before the stretching began. The changes in the lengths of the box before and after stretching and compression are shown in the **Fig. R9** below.

Fig. R9 The changes of the box sizes of system 1 (a) and system 2 (b).

(3) After 5 ns of stretching, the stretched system configuration was saved. This configuration was then used for both performing the MSD calculation and carrying out the compression simulation. In this sense, the NVT simulations can be considered as extracted from the NPT configuration without interrupting the ongoing stretching and compression process.

(4) Due to the presence of numerous small ionic liquid molecules in the simulated system, the response to stretching and compression is very fast. In fact, after the stretching is stopped, the system relaxes to equilibrium within a very short time. Furthermore, the high degree of linearity confirms that the simulation has essentially reached convergence (Fig. R10).

Fig. R10 The MSD curves of system 1 and system 2.

➤ **Reviewer #4 (Remarks to the author):**

This study presents a supramolecular polyurethane engineered with fluorine-rich segments that form electrostatic crosslinks with positively charged ionic groups at polymer chain terminals and establish fluorine-dipole interactions with blended ionic liquid to stabilize ion transport pathways. Many interesting results are presented. However, the reviewer suggests the following points to be seriously addressed.

➤ **Response #4:**

We greatly appreciate your positive assessments and constructive comments on our work. To fully address your comments and concerns, we have updated the manuscript and provided detailed explanations. Moreover, we are sincerely honored by the positive comments from the reviewers regarding our work. The main changes in the revised manuscript or supporting information are in yellow (revised contents) and blue (supplementary contents). In the following content, we will provide in-depth discussions and responses to each of your questions.

• **Comment #4-1:**

In Abstract, The author points out: “The resulting ionically conductive elastomer combines shape memory capacity, self-healing property, and exceptional cryogenic tolerance, retaining robust mechanical strength (~32.31 MPa), toughness (~107.05 MJ m⁻³) and substantial ionic conductivity even at -40 °C. Notably, it achieves an unprecedented combination of record-low operational temperatures (-40 °C to -30 °C) and an ultrahigh temperature coefficient of resistance (TCR = 8.05 % °C⁻¹), setting a new benchmark for cryogenic sensing materials.” However, the author did not focus on testing the various characteristics and application under low-temperature conditions.

• **Response #4-1:**

Thank you for your careful review and constructive suggestions on our manuscript, which greatly improved its quality. We agree that a comprehensive evaluation of the

material's performance under cryogenic conditions is crucial to support our claims in the abstract. In response to this valuable feedback, we have now significantly expanded the low-temperature testing section to provide a more robust demonstration of its cryogenic stability and applicability.

Specifically, we have supplemented the manuscript with the following key experiment:

Long-term Stability Test at -40 °C: To directly prove the exceptional cryogenic tolerance and operational stability, we conducted a continuous 120-minute resistance monitoring test at -40 °C. The results show an outstandingly minimal resistance drift of less than 4% over this extended period, which strongly confirms the material's robustness for long-term use in ultra-low temperature environments. This new data has been added to **Fig. R11 (Fig. 17 in the revised Supplementary Information)**.

Fig. R11 (Fig. 17 in the revised Supplementary Information) $\Delta R/R_0$ of PUFT2-50%IL at -40 °C for 120 minutes.

This new data, combined with the previously presented:

- (1) High ionic conductivity at -40 °C (**Fig. 5b in the manuscript**).
- (2) Ultrahigh TCR value (8.05 % °C⁻¹) in the cryogenic range (-40 to -30 °C) (**Figs. 5c-d in the manuscript**).

(3) Stable performance over five rapid thermal cycles (**Figs. 5f in the manuscript**).

These collectively forms a comprehensive suite of evidence that fully supports our claim of "exceptional cryogenic tolerance" and its suitability as a benchmark material for cryogenic sensing applications. The long-term stability test specifically addresses the concern about durability under sustained low-temperature operation, moving beyond mere functionality to proven reliability.

We have revised the relevant results section and discussion to incorporate the analysis of this new finding.

The specific changes in the manuscript are as follows:

“Furthermore, its outstanding low-temperature stability was confirmed by a minimal resistance change of less than 4% over a continuous 120-minute test at -40°C (Supplementary Fig. 17), underscoring its strong potential for reliable applications in cryogenic environments.”

- **Comment #4-2:**

In Section 2.6 (Highly sensitive ionic skin), there are too few tests on static characteristics, and basic characteristic indicators such as sensitivity, linearity and range are missing.

- **Response #4-2:**

Many thanks to the reviewer for your careful reading of our manuscript and relevant remarks. We apologize for missing these important parameters. Based on your advice, we have added experiments and descriptions in the revised manuscript. The sensitivity, linearity and range are shown in **Fig. R12 (Fig. 6a in the revised manuscript)**.

Fig. R12 (Fig. 6a in the revised manuscript) Strain sensitivity curve of PUFT₂-50%IL.

The specific changes in the manuscript are as follows:

“Meanwhile, the sensitivity of PUFT2 was evaluated under strain conditions ranging from 0 to 500%. The gauge factor (GF) was determined to be 1.93 within the 0-100% strain range and was observed to increase with increasing strain, reaching a value of 2.41 at 200-300% strain.”

- **Comment #4-3:**

In Section 2.7 (Self-healing properties), Fig. 6d-e and Fig. 6f should be marked as Fig. 7d-e and Fig. 7f. This mistake should be due to the author's carelessness.

- **Response #4-3:**

Many thanks to the reviewer for your careful reading of our manuscript and relevant remarks. We apologize for the inconvenience caused to the reviewer by the incorrect numbering. We have corrected it in the revised manuscript.

The specific changes in the manuscript are as follows:

“Healed samples (1 hour healing) successfully monitored repetitive finger and wrist flexion cycles, with $\Delta R/R_0$ amplitude consistency confirming stable real-time motion detection (**Supplementary Figs. 19-20**). Furthermore, the healed material maintained stable ECG signal acquisition capability post-repair, as evidenced by unaltered signal fidelity in **Supplementary Fig. 21**.”

- **Comment #4-4:**

As shown in Fig.5 and Fig.6, compared with the gauge factor, the resistance temperature coefficient is too strong. In the actual application process, it will affect its application in strain or pressure.

- **Response #4-4:**

We appreciate the reviewer’s insightful comment. Indeed, the high temperature coefficient of resistance ($TCR = 8.05 \% \text{ } ^\circ\text{C}^{-1}$ in the range of -40 to $-30 \text{ } ^\circ\text{C}$) indicates that the material is extremely sensitive to temperature changes, which is beneficial for cryogenic temperature sensing but may introduce thermal interference in strain or pressure sensing under fluctuating temperatures. Based on your advice, we would like to clarify the scope and operating modes evaluated in this work.

We would like to clarify that a primary objective of this work was to develop a functional material capable of reliable operation under cryogenic conditions. Therefore, its exceptional cryogenic temperature-sensing performance (exemplified by the ultra-high TCR and record-low operational temperature) is a core innovation and a deliberately pursued feature of our work, rather than an unintended side effect.

That said, we also value its strain-sensing capability. As the reviewer pointed out, the strong TCR implies that this sensor is more suitable for strain detection in environments with minimal temperature fluctuations. For instance, when attached to human skin for motion monitoring (where temperature is stable around $\sim 25\text{-}35 \text{ } ^\circ\text{C}$), the temperature interference is negligible. In response, we have supplemented the manuscript with a new linear fitting analysis of the resistance under stretching at room temperature (**Fig. 6a in the revised manuscript**), which confirms its reliable and linear

strain response under such conditions, making it fully adequate for biomechanical monitoring like human motion detection.

In conclusion, we believe this material opens a unique application niche: its primary application is as a high-performance cryogenic temperature sensor, while secondarily, it can function as a reliable strain sensor in thermally stable environments. Future work could focus on decoupling the signals through integrated reference units or compensation algorithms, but that extends beyond the scope of this manuscript as a materials study. We are grateful to the reviewer for prompting us to provide a clearer explanation of this important aspect.

- **Comment #4-5:**

It is recommended to describe the material properties first and then the application properties. For example, Section 2.7 should be described earlier.

- **Response #4-5:**

Thank you very much for your careful review and constructive suggestions regarding our manuscript. In the revision, we have reorganized the “Section Results” so that material properties precede application demonstrations. Specifically, the sections now appear in the following order:

(1) Self-Healing, damping and shape memory properties (previously Sec. 2.4)

The specific changes in the manuscript are as follows:

“Driven by reversible intermolecular multi-hydrogen bonding and ionic interactions, the PUFT₂-50%IL ionic elastomer exhibits exceptional self-healing. Real-time resistance measurements during multiple surgical blade cuts showed that the material restored its original resistance within 0.5 seconds, confirming rapid electrical recovery (Fig. 4a). To quantify self-healing efficiency, PUFT₂-50%IL samples were cut

into two pieces, then the two completely separated ruptured surfaces were brought into full contact at room temperature. As shown in Fig. 4b, tensile strength and strain gradually increased with healing time. After 1 hour, the tensile strength and strain recovered to 0.74 MPa and 760%, respectively. After 3 hours, they reached 1.11 MPa and 912%, similar to the original sample. For strain sensors, stretchability is a critical parameter. Based on strain recovery, the healing efficiency reached 85.4% after 1 hour and 97.4% after 3 hours (Fig. 4c), indicating nearly ideal self-healing performance.”

(2) Low temperature tolerance and temperature sensitivity (previously Sec. 2.5)

(3) Highly sensitive and healing ionic skin (previously Sec. 2.6)

The specific changes in the manuscript are as follows:

“Previously, the excellent self-healing performance of PUFT₂-50%IL had already been confirmed. Therefore, it could be inferred that PUFT₂-50%IL would still be able to maintain its sensing ability after healing. Healed samples (1 hour healing) successfully monitored repetitive finger and wrist flexion cycles, with $\Delta R/R_0$ amplitude consistency confirming stable real-time motion detection (Fig. 7d–e). Furthermore, the healed material maintained stable ECG signal acquisition capability post-repair, as evidenced by unaltered signal fidelity in Fig. 7f. These results collectively demonstrated that the self-healing mechanism preserved both mechanical integrity and sensing functionality, significantly extending the material’s applicability in demanding environments. This work substantiated that hierarchical dynamic bonding networks not only enabled robust self-healing but also ensured performance retention across mechanical, electrical, and bio-signal sensing modalities, establishing PUFT₂-50%IL

as a durable platform for next-generation adaptive sensors.”

This reordering separates intrinsic material behaviors (temperature response, healing) from application figures, while preserving the key message that healed devices remain fully functional for sensing. We hope this addresses the reviewer’s concern.

- **Comment #4-6:**

Generally, ionic conductors are commonly used in capacitive sensors, demonstrating high sensitivity, but are less applied in resistive principles. It can also be roughly seen from the results in Section 2.6 (Highly sensitive ionic skin) that the gauge factor of the designed sensor is relatively low.

- **Response #4-6:**

Thanks for your careful review and constructive suggestions regarding our manuscript. We fully agree that ionic conductors are predominantly employed in capacitive sensing modes due to their ability to achieve high sensitivity, particularly for detecting minute pressures or strains.

However, the primary objective of this part of our work was to explore the often-overlooked potential of ionic conductors in resistive sensing schemes. We intentionally designed our sensor for applications that require large-scale deformations (e.g., >200% strain, such as in joint movement monitoring or soft robotics). To thoroughly demonstrate this advantage, we have now supplemented the sensitivity data under 0-500% strain in the revised manuscript, as kindly suggested by the reviewer (**Fig. R13**). The new results clearly show that our sensor exhibits a stable and monotonic increase in resistance with strain, with a GF of 1.93 in the 0-100% range, increasing to 2.41 at 200-300% strain. Most importantly, it maintains excellent linearity and reliability even up to 500% strain, a performance that is challenging to achieve with many capacitive sensors requiring fixed parallel plates.

Therefore, we posit that the value of our work lies not in competing with capacitive ionic sensors for high sensitivity at small strains, but in demonstrating a viable and

robust resistive-type ionic skin that is uniquely suited for high-strain applications. This expands the application repertoire of ionic conductors beyond the conventional capacitive paradigm.

Fig. R13 (Fig. 6a in the revised manuscript) Strain sensitivity curve of PUFT₂-50%IL.

- **Comment #4-7:**

The application part of the material is too simple to support its application value.

- **Response #4-7:**

First of all, I would like to thank the reviewer for pointing out this problem. We agree that demonstrating tangible application value is crucial. In our study, we aimed to highlight the application potential not through a single complex device, but by demonstrating a suite of advanced material properties that, when combined, directly address critical limitations in the field of soft electronics and wearable technology.

Specifically, the value of our material platform lies in its multifunctionality and enhanced durability, which are key for practical applications:

- (1) Its core functionality as a highly sensitive ionic skin for wearable motion

sensing (Figs. 4d-h in the revised manuscript) and temperature sensing (Fig. 4c in the revised manuscript) establishes its primary application domain.

(2) The exceptional low-temperature tolerance (Fig. 4f in the revised manuscript) significantly expands its application scenarios from room condition to harsh environments (e.g., winter sports, polar exploration), a domain where most hydrogel-based devices fail.

(3) The self-healing capability and full recovery of sensing performance post-healing (Figs. 4a-c and Supplementary Figs. 17-19) are not merely demonstrations but are central to its application value. They directly solve the problem of mechanical failure and short lifetime, promising reduced maintenance costs and longer service hours for wearable devices.

(4) The shape memory property (Fig. 4g) further adds a dimension for intelligent applications, such as pre-programmed fitting or actuation.

To better integrate these points and address the reviewer's concern, we have now added the resistance changes at -40°C for two hours to prove stability (Fig. R14).

Fig. R14 (Fig.17 in the revised Supplementary Information) $\Delta R/R_0$ of PUFT₂-50%IL at -40°C for 120 minutes.

In the end, thank you again for prompting us to better articulate this point.

Response to Previous Reviewers' Comments

Dear reviewers,

We would like to thank all reviewers for their constructive comments and suggestions. These comments are valuable and helpful for the improvement of our paper, and they also have important guiding significance for research. Therefore, we have studied the comments carefully and have made further revisions, hoping that the reviewers and editors will be satisfied with the revised manuscript.

Our point-by-point responses to these comments are provided below, and the main changes in the revised manuscript are marked in yellow (revised contents) and blue (supplementary contents). In any case, we are open to consideration of any further comments and suggestions. We look forward to hearing from you on the status of our submission.

Sincerely,

Xinrui Zhang and Jing Xu

➤ **Reviewer #3 (Remarks to the author):**

This review evaluates the molecular modeling aspects of the revised manuscript. I now consider the manuscript acceptable for publication but I would appreciate if the authors could address the following comments:

➤ **Response #3:**

Thank you for your affirmation and reviewing the manuscript in your busy schedule and provided us with your professional comments and suggestions, the questions and the certain responses were presented in following part orderly.

• **Comment #3-1:**

In the rebuttal the authors say "Structural optimization is initially carried out by adjusting the atomic positions based on the direction of atomic forces, moving them toward a lower total potential energy. This process is done to avoid excessively high initial energy of the structure." This is fine if the sole purpose is to avoid high initial energy structures. But this local relaxation will not cause an extended chain to become coiled or vice versa. Probably "local relaxation" is a better term here than optimization.

• **Response #3-1:**

We sincerely thank you for your insightful comments and have thoughtfully revised the manuscript based on the reviewers' comments. We have corrected it in the revised Supplementary Information.

The specific changes in the Supplementary Information are as follows:

“Initially, the simulation systems underwent structural local relaxation.”

• **Comment #3-2:**

Fig. R6 is helpful so I can see what the full system is. Thank you for including this. However, in the rebuttal the authors say " Driven by thermal motion, the polymer chains

spontaneously bent and entangled, forming the structure corresponding to ambient temperature and pressure." It is far from obvious that any entanglement occurs. Can the authors substantiate this with a measurable quantity, in other words can they quantify entanglement?

- **Response #3-2:**

We thank the reviewer for raising this critical point. We agree that the term "entanglement" may seem qualitative without rigorous quantitative support. In our context, we used it to describe the process whereby polymer chains form a stable and natural network structure. We tracked the evolution of the radius of gyration of the polymer chains. The observed decrease and eventual stabilization of the radius of gyration indicate a transition from a relatively extended initial conformation to a more compact one, consistent with the chains rearranging through interactions to form a stable structure.

To enhance accuracy, we have modified the description to specify that "**the chains underwent relaxation, reaching a stable radius of gyration,**" as this more precisely conforms to the conformational state we measured.

The specific changes in the manuscript are as follows:

“The time evolution of the average radius of gyration for the polymer molecules, [EMIM]⁺, and [TFSI]⁻ was analyzed during the relaxation process to confirm that the system had reached equilibrium. ”

- **Comment #3-3:**

In Fig. R7 the authors show the MSD vs. time plot which I asked for. This is appreciated. However, since the diffusion constant is quite low (4 orders of magnitude lower than water) it is not convincing to stop the plots at 2 ns (= 2000 ps). The data sort of looks linear but it would be much more convincing if the time scale was extended.

- **Response #3-3:**

Thanks again for your careful review and constructive suggestions on our manuscript, which greatly improved its quality. In accordance with the reviewer's suggestion, we have extended the MSD simulation time to 15 ns. Concurrently, the time span for MSD calculation was increased to 10 ns. The resulting MSD curves for the different ion types are shown in **Fig. R1**, which demonstrate good linearity.

Fig. R1 The MSD curves of system 1 and system 2.

- **Comment #3-4:**

In the rebuttal the authors say "We define the steady state as the condition in which the volume of the simulation box no longer changes and the total energy has converged." These are very weak criteria for equilibrium. More convincing would be to track the radius of gyration of the polymer chains. This is because changes in the radius of gyration or other polymer conformational properties might not show up in the overall system volume and would be a more careful test of steady state.

- **Response #3-4:**

We sincerely thank you for your kind suggestions. Following the reviewer's suggestion, we tracked the radius of gyration of the polymer chains, which rapidly converged to a stable value (**Fig. R2**), confirming the system had reached equilibrium.

This new stability criterion has been detailed accordingly in the revised Supplementary Information.

Fig. R2 The radius of gyration of system 1 and system 2.

The specific changes in the manuscript are as follows:

“The time evolution of the average radius of gyration for the polymer molecules, [EMIM]⁺, and [TFSI]⁻ was analyzed during the relaxation process to confirm that the system had reached equilibrium.”

• **Comment #3-5:**

In the rebuttal the authors say "After compression, the lengths in the y and z directions return to the values observed before the stretching began". Were the y and z box lengths constrained to have the same value as each other? Otherwise it would be surprising for the y and z values to return exactly to their starting values. If such a constraint was used it should be stated.

• **Response #3-5:**

Thank you very much for your careful review and constructive suggestions regarding our manuscript. We apologize for the lack of this information. Yes, this

constraint is indeed in place because we aim to simulate uniaxial tensile loading. To maintain stable box dimensions, we constrained the dimensions of y and z directions to remain consistent. We have now added this detail in the revised Supplementary Information.

The specific changes in the Supplementary Information are as follows:

“Following this, over a period of 5 nanoseconds, the systems were uniformly stretched in the x direction to twice their original length, while maintaining a pressure of 1 atmosphere in the y and z directions. To maintain stable box dimensions, we constrained the dimensions of y and z directions to remain consistent. Then, over another 5 nanoseconds, the systems were compressed back to their original length in the x-direction...”